# Spatial and Temporal Variability of Rainfall Trends in Response to Climate Change—A Case Study: Syria

**Martina Zeleňáková** [1,*], **Hany F. Abd-Elhamid** [2,3], **Katarína Krajníková** [4,*], **Jana Smetanková** [3], **Pavol Purcz** [4] **and Ibrahim Alkhalaf** [1]

1   Environmental Engineering Department, Faculty of Civil Engineering, Technical University of Kosice, Vysokoškolská 4, 042 00 Kosice, Slovakia; alkhalaf.ibrahim@gmail.com
2   Department of Water and Water Structures Engineering, Faculty of Engineering, Zagazig University, Zagazig 44519, Egypt; hany_farhat2003@yahoo.com
3   Center for Research and Innovation in Construction, Faculty of Civil Engineering, Technical University of Košice, 042 00 Košice, Slovakia; jana.smetankova@tuke.sk
4   Department of Applied Mathematics and Descriptive Geometry, Faculty of Civil Engineering, Technical University of Kosice, 040 01 Košice, Slovakia; pavol.purcz@tuke.sk
*   Correspondence: martina.zelenakova@tuke.sk (M.Z.); katarina.krajnikova@tuke.sk (K.K.); Tel.: +421-55-602-4270 (M.Z.)

**Abstract:** Recent climate changes have prompted changes in the hydrological cycle at a global scale, creating instability when predicting future climate conditions and related changes. Perturbations in global climate models have created the need to concentrate consequent changes in hydro climatic factors to comprehend the regional and territorial impacts of climate and environmental changes. Syria, as a Middle East country, is exposed to extreme climate events such as drought and flood. The aim of this study is to analyze rainfall trends in Syria in response to the likely climate change. The analysis was conducted for rainfall data collected from 71 stations distributed all over the country for the period (1991–2009). The trend analysis was performed in monthly and seasonal scales using Mann–Kendall non-parametric statistical tests. The results attained from Mann–Kendall trend analysis revealed decreasing trends at most of the stations. Additionally, rainfall analysis was conducted for the stations with significant trends for wet and dry periods, which also revealed decreasing trends at almost all the stations. From the analysis of the results, it is obvious that slight increasing trends in rainfall in Syria occurred in the fall period. However, in the winter and spring periods, significant decreasing trends have been observed at almost all the stations. This reveals that the country will suffer from shortage of water, because most rainfall occurs in the winter and spring, infrequently in fall and rarely in summer. The results are consistent with the IPCC's fifth report that predicted a decrease in rainfall in the Mediterranean and southern Asia. The results of this paper could help the management of water resources in Syria considering future climate changes.

**Keywords:** climate change; rainfall trend analysis; Mann–Kendall; water scarcity; Syria

## 1. Introduction

A shift in average weather conditions, or the time variation of weather in the longer-term, is referred to climate change (e.g., more or fewer extreme weather events) [1]. A number of mechanisms contribute to climate change including biotic activities, fluctuations in solar energy received by Earth, plate tectonics, and volcanic eruptions. Certain human activities have also been identified as main contributors to current climate change [1]. Global warming is projected to have far-reaching and long-term consequences. Human activities, particularly the combustion of fossil fuels, causes the accumulation of carbon dioxide, methane, and other greenhouse gases in the atmosphere, causing the steady warming of the Earth's surface, oceans, and atmosphere [2]. Global warming is a component of climate change in terms of the long-term rise in global temperatures. The major cause of global

warming is the greenhouse gases. They include carbon dioxide, methane, nitrous oxides, and, in some cases, chlorine and bromine containing compounds. Their overall effect is to warm the Earth's surface and the lower atmosphere, because greenhouse gases absorb some of the outgoing radiation of Earth and re-radiate it back towards the surface [3]. As a result of climate change, the occurrence of extreme weather events has increased. Based on numerous data, it is now more certain than ever that humans are changing the Earth's climate. Warming of the atmosphere and oceans has been followed by rising sea levels, a significant reduction in Arctic sea ice, and other climate-related phenomena. Climate change's effects on humans and wildlife are becoming more obvious. Flooding, heat waves, and wildfires have caused billions of dollars in damage. In reaction to changing temperatures and precipitation patterns, habitats are rapidly shifting [4,5].

Climate pressures, or "enforcement mechanisms", are factors that influence the climate. Fluctuations in the Earth's orbit, variations in solar radiation, variations in the albedo or reflectivity of the continents, atmosphere, and oceans, mountain-building and continental drift, and changes in greenhouse gas concentrations are all examples of these processes [6–8]. Recent studies have revealed that global warming is currently occurring as a result of human activities. If current trends continue, global warming would likely reach 1.5 °C between 2030 and 2052. Over the next 100 years (2000–2100), the global average surface temperature is expected to rise from 2.6 °C to 4.8 °C under RCP 8.5 (high emission scenario) and from 0.3 °C to 1.7 °C at RCP 2.6 (low emission scenario) (See Figure 1). As a result of the uncertainty measure and time series of projections, the global average temperature will most likely rise in the future [9–11].

Changes in water temperature in rivers and lakes are likely to be the most rapid response to climate change. Water temperature rises in tandem with the temperature of air. Higher temperatures, according to recent studies, boost microbial activity in the sediments and soil at the bottom of lakes and rivers, speeding up the release rate of internal Phosphorus (P) loading, which could account for a significant amount of the overall nutrient load in the water [12]. Furthermore, a warmer temperature reduces surface water viscosity and increases nutrient diffusion to the cell surface, which is an important mechanism when species compete for nutrients. According to recent studies, increasing air temperature causes the eutrophication process in water bodies to speed up by changing the water temperature (even if the external source of nutrients has become stable). As a result, cyanobacteria will be more likely to dominate phytoplankton assemblages in eutrophic freshwater habitats, particularly in temperate ecosystems, during the hottest times of the year [12,13].

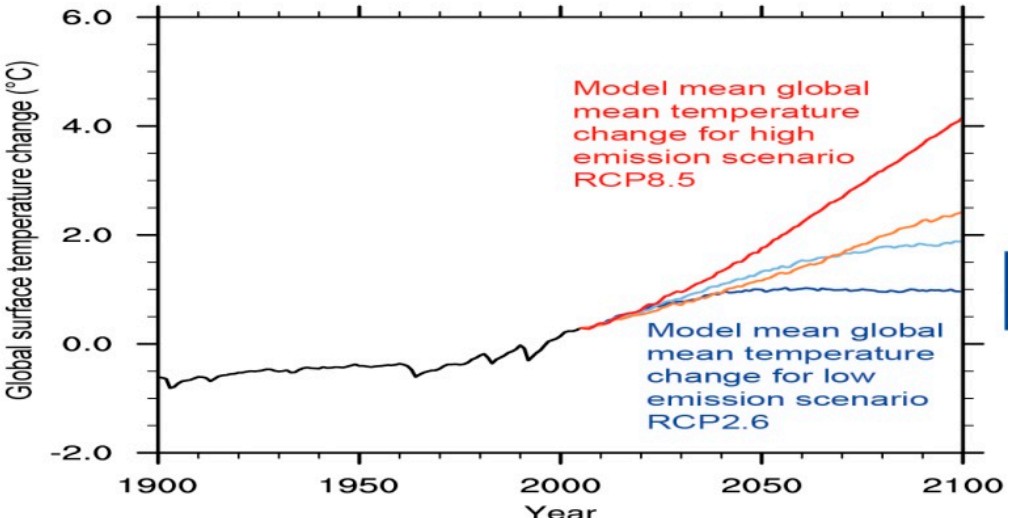

**Figure 1.** Changes in global surface temperature between 2000 and 2100 [14].

Scientists are paying more attention to the wind as a crucial contributor to climate change. There is evidence of long-term changes in large-scale atmospheric circulation due to variations in wind direction and strength. Stronger wind speeds will be observed in the boreal parts of the northern hemisphere by 2050, according to simulations, including Canada, tropical and subtropical regions, northern Europe, and central and southern America. In southern Europe, eastern and southern Asia, and much of the western coast of South America, the opposite effect will be observed, a reduction in wind speed [15]. The effects of wind on water eutrophication are both direct and indirect. Wind's immediate consequences are algae blown from the lake to the lake shore, causing algal blooms that affect residents along the lake or river. The wind's indirect effect is an increase in waves, which speeds up the release of nutrients from sediments in the water. Higher wind speeds, according to research, impact water movement and circulation, reducing water stability and boosting nutrient mixing [16].

In addition to temperature and wind effects, precipitation-induced changes in hydrological regimes are also important. While global temperature trends are rising, precipitation variations will be uneven [17]. Results show that annual mean precipitation will increase around the equatorial Pacific and some high-latitude areas, especially under the RCP 8.5 scenario, as shown in Figure 2, based on multi-model mean projections for 2081–2100 compared to 1986–2005 under the RCP 2.6 (left) and RCP 8.5 (right) scenarios. However, mean precipitation is anticipated to drop in several mid-latitude and subtropical dry regions, while mean precipitation is expected to increase in many mid-latitude regions under the same scenario. As a result, there is still a chance that extreme precipitation events will become more intense and frequent over most mid-latitude land masses and wet tropical regions, where the water will continue to warm and acidify and the global mean water level will rise [18]. Hydraulic features, flow rate, water level, inundation pattern, and water cycle are all predicted to be affected by precipitation. The probability of hydrological extremes such as drought and flooding occurrences will rise, especially if hydrological regimes change due to intense precipitation. Panagoulia (1995) [19] presented the assessment of daily catchment precipitation in mountainous regions for climate change interpretation in Mesochora catchment in Greece.

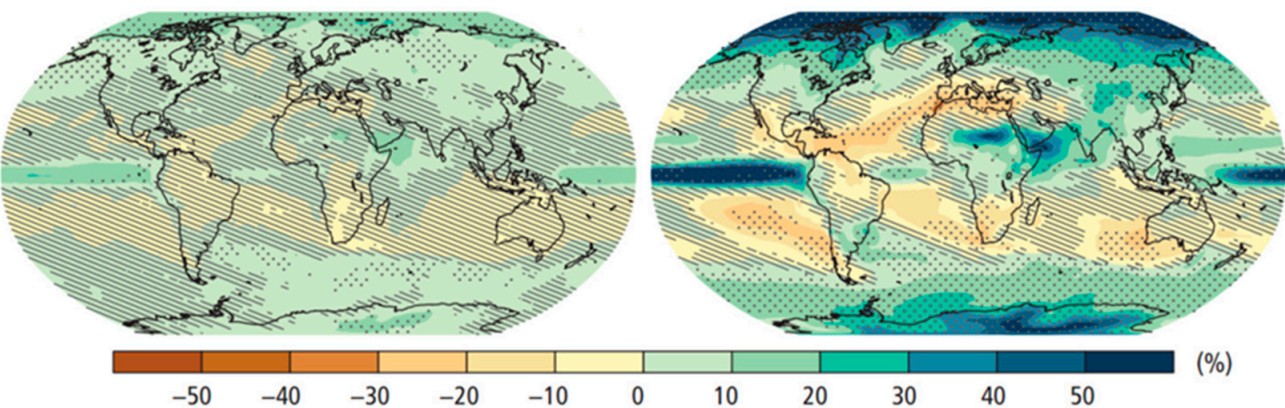

**Figure 2.** Average precipitation changes from 1986 to 2005 and 2081 to 2100 [14].

As a result of climate change, the intensity and distribution of temperature, wind, and precipitation on the globe may alter. According to the studies shown above, climate change will have an impact on hydrological regimes, increasing the probability of hydrological extremes such as drought and flooding. The Middle East is one of the most climate-vulnerable regions on the planet. It is regarded as the world's most difficult region. Additionally, due to the possibility of future conflicts over water resources, the region has been categorized as a frights region. Most Middle Eastern countries are experiencing critical water shortages. Syria is a Middle Eastern country that can serve as an indicator for the region. Syria is primarily comprised of semi-arid, arid, and hyperaridic regions. The region is marked by

rapid population growth, which has resulted in a substantial drop in per capita water share in recent decades.

Over the years, a number of studies have been conducted for rainfall trend analysis around the world using different methods. Haylock et al., (2006) [20] examined the daily rainfall records to determine changes in both total and extreme rainfall from 1960 to 2000 in the region of southern Brazil, Paraguay, Uruguay, and northern and central Argentina. A decrease was observed in southern Peru and southern Chile, with the latter showing significant decreases in many indices. A significant decrease in many of the rainfall indices at several stations in southern Chile and Argentina can be explained by a canonical pattern reflecting a weakening of the continental trough leading to a southward shift in storm tracks. Endo et al., (2009) [21] investigated the precipitation trends using daily data from Southeast Asian countries from 1950 to 2000. The results reveal that the number of stations with significant upward trend is larger than that with significant downward trend. Alahacoon and Edirisinghe (2021) [22] presented spatial variability of rainfall trends in Sri Lanka from 1989 to 2019 as an indication of climate change. Sen's slope estimator and the Mann–Kendall test were used to investigate the trends in annual and seasonal rainfall throughout all districts and climatic zones of Sri Lanka. The results showed a significant increase in annual rainfall from 1989 to 2019 in all climatic zones of Sri Lanka. Sabattini et al., (2021) [23] evaluated the long-term changes in the intensity of rainfall in the central-north region of Entre Ríos between 1945 and 2019, based on daily precipitation records aggregated at yearly, monthly, and seasonal levels. Alahacoon et al., (2022) [24] investigated the rainfall variability and trends in the African continent using data from 1983 to 2020. The Mann–Kendall test and Sen's slope estimator were used to analyze rainfall trends and their magnitude under monthly, seasonal, and annual time scales.

Some studies have considered specific water challenges in Syria. Burdon and Safadi [25] investigated the geological formation of Syria's groundwater aquifers. The long-term management of water in Hawran, southern Syria, was studied by Braemer et al. [26]. Altonbilek [16] investigated the Euphrates and Tigris basin's development and administration. The hydrological and watershed parameters of the El-Kabir River between Syria and Lebanon were evaluated by Shaaban et al. [27]. Kattan [28] studied the hydrological and environmental features of surface and groundwater in the Barada and Awaj basins in Damascus. Only a few studies have examined Syria's overall water resources [29,30]. Furthermore, the impact of climate change on water resources was not taken into account in prior research. Another key issue in Syria is the impact of climate change on rainfall pattern. Rainfall is a key meteorological quantity that must be accurately recorded and assessed. This study will analyze rainfall trends in Syria as a result of likely climate change in order to determine how much water is available to meet various demands. Data from 71 stations was analyzed for the period (1991–2009). The data was gathered from Aleppo University in Syria, with the help of the Syrian Ministry of Agriculture and Agrarian Reform and the Syrian Meteorological Centre.

## 2. Study Area and Data Preparation

### 2.1. Study Area

Syria is located in western Asia, north of the Arabian Peninsula, on the Mediterranean Sea's eastern shore. Turkey, Lebanon, Israel, Iraq, and Jordan are its neighbors (see Figure 3). Syria has mountains in the west and a steppe area in the east. Mount Hermon of height (2814 m) on the Syrian–Lebanese border is the highest level in Syria. A semi-arid steppe zone spanning three-quarters of the country, which experiences hot, dry winds blowing over the desert, situated between the humid Mediterranean coast and the arid desert regions. Syria is severely depleted, with only 3% forest and woodland, 28% arable land, 4% permanent crops, and 46% meadows and pastures. Syria's deserts, plains, and mountains cover an area of 185.180 km$^2$. It is separated into two sections: a coastal zone with a thin, double mountain belt enclosing a depression in the west and a considerably bigger plateau in the east [31]. The climate is primarily arid, with less than 250 mm of rain

falling on around three-fifths of the land each year. The difference is the most remarkable characteristic of the climate. The majority of the rain falls between November and May. The average yearly temperature is 7 °C in January and 27 °C in August [32,33]. Furthermore, the light rainfall is highly changeable from year to year, resulting in recurrent droughts. In July, temperatures in the barren rocky desert south of Jabal al Ruwaq, Jabal Abu Rujmayn, and Jabal Bishri mountains frequently surpass 45 °C [34].

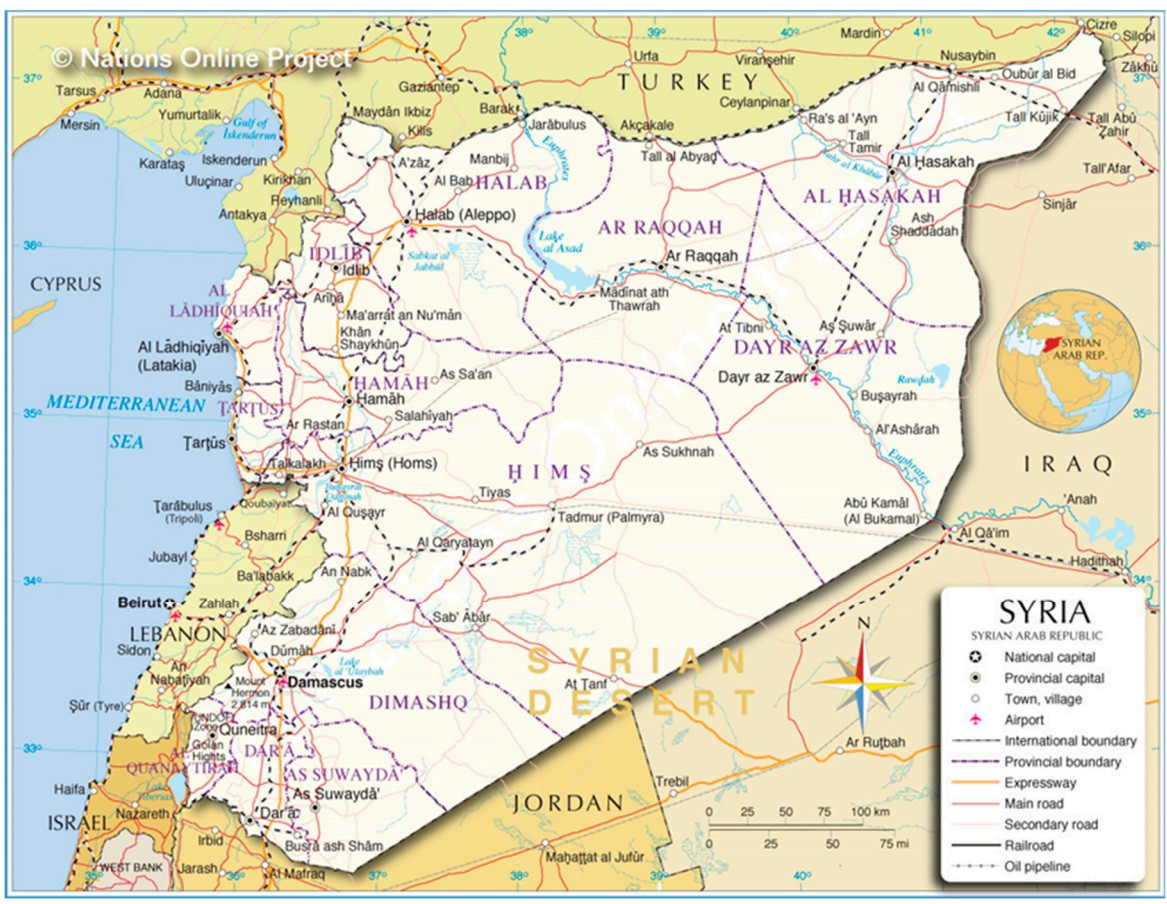

**Figure 3.** Location mag of the study area.

Syria is primarily comprised of semi-arid, arid, and hyperaridic regions. Drought and desertification cycles have recently wreaked havoc on the semiarid zone. In terms of socioeconomics, the region is characterized by a rapidly growing population, which has resulted in a severe drop in per capita water share in recent years. Water is the most critical factor for a sustainable ecosystem and, even more importantly, for economic development in these circumstances [35–37]. Furthermore, water management has a major impact on agriculture. Agriculture is the major user of water resources. Water consumption is also rising as a result of coastal development and tourism. A prediction of water scarcity is expected based on these factors. Rivers and rainfall are Syria's main sources of water, both of which are increasingly impacted by climate change. The heavy use of groundwater by agriculture over the last decade has resulted in aquifer depletion, a drop in the groundwater table, and a significant reduction in spring production. The amount of groundwater in the Syrian part of the Asi-Orontes Basin was estimated to be 1607 million m$^3$; the majority (1134 million m$^3$) runs as springs, while (473 million m$^3$) is stored in aquifers and extracted by wells for irrigation and water supply [38].

*2.2. Rainfall Data*

Syria, with a total population of 21.13 million and an area of 185,180 km², includes five agro-ecological zones that are dependent on rainfall. Humid zones can be found along the Mediterranean coast on the west. The east, north, and south have arid and semi-arid zones. Due to the predominant rainfall happening in the winter from December to March, there is a substantial seasonal variance in water resource availability. Syria's yearly rainfall drops from roughly 900 mm at the coast to about 60 mm in the eastern parts. More than 60% of the country receives less than 250 mm of rain per year, making water scarce. In the western sections of Syria, the potential evaporation rate is around 1300 mm/year, whereas in the eastern and south-eastern parts it approaches 3000 mm/year. Syria is a Middle Eastern country located between Lebanon and Turkey on the Mediterranean Sea. In the west, there is a narrow coastal plain with a double mountain belt; in the east, there is a huge semiarid and desert plateau. Summer is hot, dry, and sunny (from June to August), with mild, rainy winters (December to February) along the shore. In this study, data from 71 stations were gathered with the help of the Syrian Ministry of Agriculture and Agrarian Reform and the Syrian Meteorological Centre, Aleppo University in Syria. The data was analyzed for the period from 1991 to 2009. Figure 4 shows the locations of 71 rainfall gauge stations, which are located around the country.

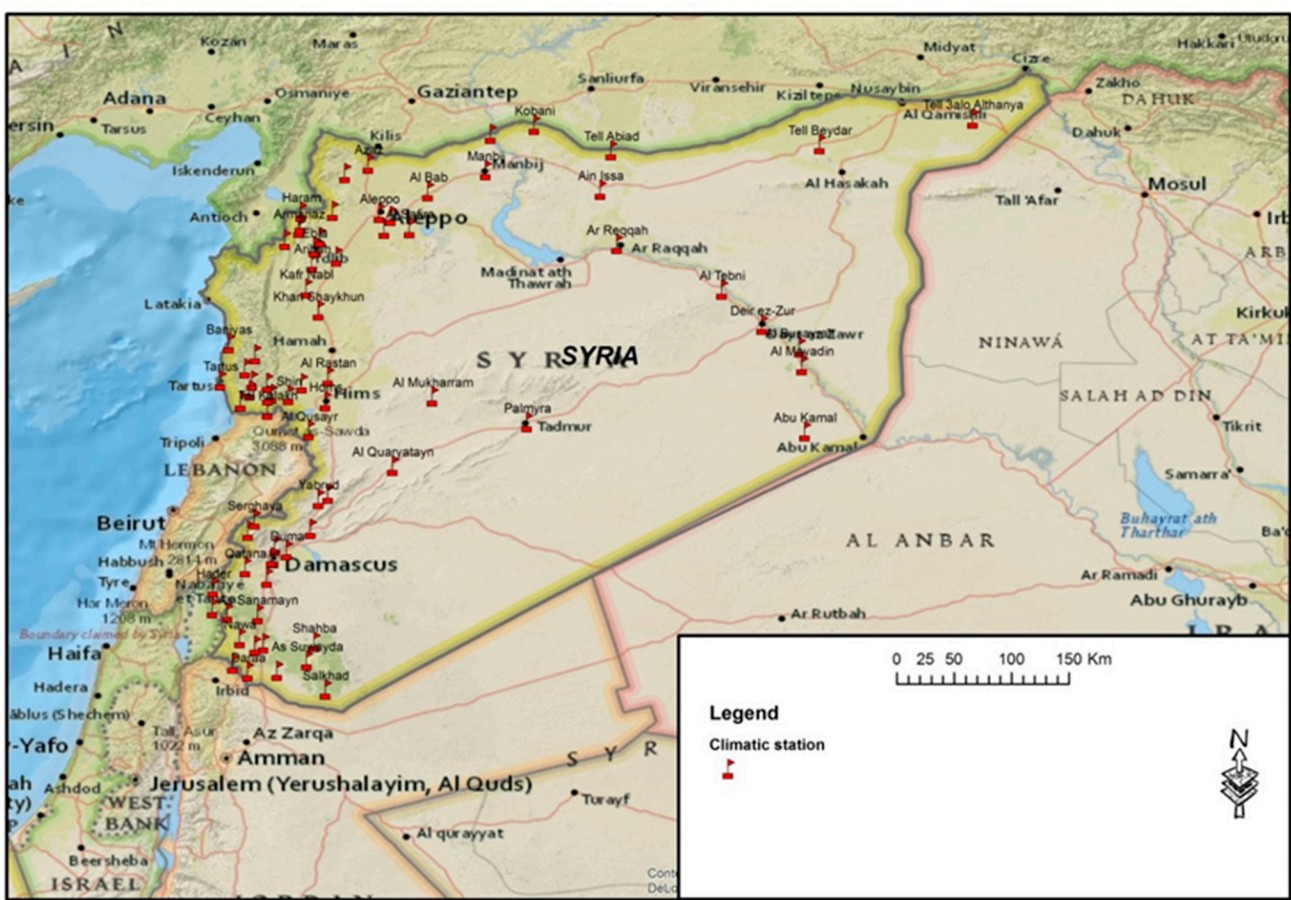

**Figure 4.** Location of the 71 rain gauges stations in Syria.

This study analyzed data from 71 stations. The average annual rainfall at the 71 stations from 1991 to 2009 is shown in Figure 5. The maximum annual rainfall reached 1200 mm and the minimum is 100 mm (Figure 5).

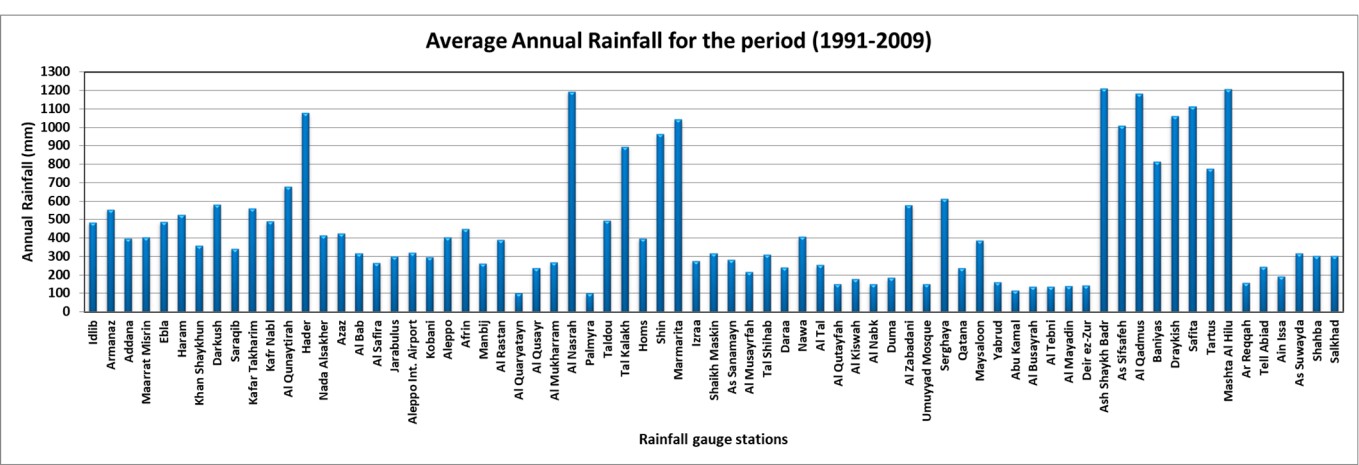

**Figure 5.** Annual rainfall at 71 rain gauges' station in Syria.

## 3. Methodology

Based on the above results, this study aims to analyze trends in the amount of precipitation and to perform an analysis of indices in selected areas. This study examines precipitation trends in Syria, specifically in the Mediterranean area. Rainfall is a key meteorological parameter that must be carefully recorded and assessed. Many applications of rainfall data can be examined in depth with a thorough understanding of rainfall distribution. Furthermore, the amount of rainfall received across a given area is key in determining the amount of water available to meet diverse demands for agriculture, industry, and other human activities.

There is a substantial change in hydrological parameters, such as evaporation and precipitation, as a result of the large increase in world average temperature, which has a cumulative influence on river flow regimes. The data were analyzed in this study utilizing the most used statistical approaches to trend detection, such as:

- For trend detection, slope-based tests such as least squares linear regression (referred to as LR) and Sen's robust slope estimator (referred to as SS) were utilized (Sen, 1968) [39].
- For trend detection, rank-based tests such as Mann–Kendall (referred to as MK) and Spearman rank correlation (referred to as SRC) were utilized [40].

These are some of the most frequently used statistical tests for detecting hydrological trends in the literature. LR is a parametric strategy that requires both distributional and independent assumptions to be met, whereas rank-based approaches are nonparametric and only require the second assumption to be met. SS is not a statistical test in the traditional sense. A resampling strategy must be used to determine the upper and lower limits of the test statistics. In this investigation, Mann–Kendall is utilized to find trends [40–42].

**Mann–Kendall trend analysis**

The Mann–Kendall trend test is a non-parametric analysis that is the most frequent statistical approach for analyzing time series datasets. Deriving the distribution of the test statistic was first developed by Mann (1945) [43] and completed by Kendall (1975) [44]. When testing data that deviate from "normality", non-parametric techniques are a more reliable option. In addition, nonparametric approaches are supposed to be more resistant to outliers. The null hypothesis $H_0$ of the Mann–Kendall trend test is that there is no trend, and that the data are random and independent; the alternate hypothesis $H_1$ is that the time series has a trend. The statistics for the Mann–Kendall test are based on the conventional normal distribution (Z) as following [43]:

$$Z = \begin{cases} \frac{S-1}{\sqrt{\text{Var}(S)}} & \text{if} & S > 0 \\ 0 & \text{if} & S = 0 \\ \frac{S+1}{\sqrt{\text{Var}(S)}} & \text{if} & S < 0 \end{cases} \tag{1}$$

in which

$$S = \sum_{k=1}^{n-1} \sum_{j=k+1}^{n} \mathrm{sgn}(x_j - x_k) \tag{2}$$

$$\mathrm{sgn}\left(x_j - x_k\right) = \begin{cases} +1 & \text{if } \left(x_j - x_k\right) > 0 \\ 0 & \text{if } \left(x_j - x_k\right) = 0 \\ -1 & \text{if } \left(x_j - x_k\right) < 0 \end{cases} \tag{3}$$

$$\mathrm{Var}(S) = \left[ n(n-1)(2n+5) - \sum_{i=1}^{m} t_i(t_i - 1)(2t_i + 5) \right] / 18 \tag{4}$$

where n is the number of points in the data set and m is the number of entangled groupings (a set of sample data with the same value).

The null hypothesis $H_0$ states that the depersonalized data $(x_1, \ldots, x_n)$ is a sample of n independent and identically distributed random variables. The alternative hypothesis $H_1$ of a two-sided test is that the distributions of $x_k$ and $x_j$ are not identical for all $(k, j \le n)$ with $(k \ne j)$. The significance level is chosen as $\alpha = 0.05$ and $Z_\alpha/2$ is the value of normal distribution function that taken $(Z_\alpha/2 = 1.645)$ in this study. Hypothesis $H_0$ is no trend if $(Z < Z_\alpha/2)$, and $H_1$ is there is a trend if $(Z > Z_\alpha/2)$. Positive values of Z indicate increasing trends, while negative values of Z show decreasing trends. If $(n \ge 10)$, the test statistic can be approximated using normal distribution, and the normalized test statistic Z can then be computed. In two-tailed test, $(|Z| > Z_\alpha/2)$ indicates that the null hypothesis has been rejected, with $\alpha$ being the significance level of the test. For this study, significance levels of 0.05 is used. The non-parametric estimate of the trend magnitude of the slope, $\beta$ of linear trend, was taken to be the Theil–Sen slope as proposed by Theil and Sen (Sen, 1968) [30]. The Theil–Sen slope $(\beta)$ is calculated as the median of all possible slopes:

$$\beta = \mathrm{Median}\left( (x_j - x_k) / (j - k) \right) \tag{5}$$

for i = 1, 2, ... , N, where $x_j$ and $x_k$ are data values at time j and k $(j > k)$, respectively, and N is a number of all pairs $x_j$ and $x_k$. A positive value of $\beta$ indicates an increasing trend, and a negative value indicates a decreasing trend in the time series.

## 4. Results and Discussion

This study investigates changes of rainfall trends over all Syria in the period (1991–2009). The rainfall data from 71 stations distributed in Syria is analyzed in this section. The location of the rainfall stations was shown in Figure 4 and the average annual rainfall for the stations and annual rainfall was shown in Figure 5.

### 4.1. Rainfall Trends in Syria

Rainfall is a crucial characteristic for any vulnerability assessment in a changing climate, and variations in its mean, maximum, and lowest values are used to analyze it. Therefore, the details of the seasonal mean, maximum, and minimum rainfalls are taken into account. Analyzing mean rainfall is vital for every region, but determining maximum and lowest rainfall is significant for many application-oriented tasks such as hydrological extremes, water management, and agriculture.

In this analysis, two seasons were evaluated separately as the wet season (October–January) and dry season (February–May). However, the months June, July, and August were excluded because there is no rainfall in the summer. The trend analysis was conducted for the period (1991–2009). Then, the significant stations shown in Table 1 were analyzed as follows:

**Table 1.** Seasonal trends of rainfall in significant stations in Syria for the period (1991–2009).

| | | Period | |
|---|---|---|---|
| Station | September–November | Station | September–November |
| Khan Shaykhun | 0.1552 | −0.0520 | −0.2973 |
| Al Quaryatayn | 0.0000 | 0.0020 | −0.0100 |
| Tal Kalakh | 0.7228 | 1.0171 | −0.1481 |
| Al Nabk | −0.1667 | 0.0000 | −0.0500 |
| Al Busayrah | 0.0000 | −0.1600 | −0.0950 |
| Al Tebni | 0.0029 | −0.0385 | −0.0780 |
| Ebla | 0.1460 | 0.0759 | −0.3556 |
| Al Qadmus | 0.8182 | −0.2553 | −0.7353 |
| Baniyas | 0.4545 | −0.3263 | −0.2192 |
| Ar Reqqah | 0.0000 | −0.2613 | −0.1120 |
| Ain Issa | 0.0000 | −0.3143 | −0.0308 |

The analysis of rainfall trends for the period (1991–2009) at Khan Shaykhun is shown in Figure 6. In the wet season, the highest rainfall rate was recorded in 2002 (515 mm), and the lowest rate was recorded in 2007 (221 mm). During the wet season, the rainfall has a decreasing trend (Figure 6a). However, in the dry season, the highest rainfall rate was recorded in 2005 (15 mm) and the lowest rate was recorded in 1991–1996 (0 mm). The rainfall has an increasing trend in the dry season (Figure 6b).

For Al Quaryayayn station, the analysis of rainfall trends is shown in Figure 7. The highest rainfall rate in the wet season was recorded in 2006 (171 mm) and the lowest rate was recorded in 2007 (37.9 mm). In the dry season, the highest rainfall rate was recorded in 1991 (29 mm) and the lowest rainfall rate was recorded in 1999–2008 (0 mm). The rainfall has a decreasing trend in both wet and dry seasons (Figure 7a,b).

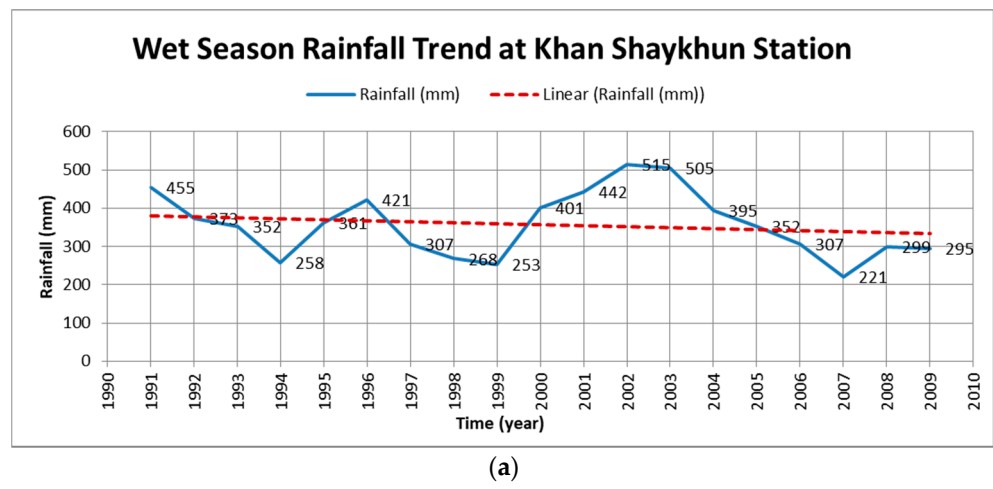

**(a)**

**Figure 6.** *Cont.*

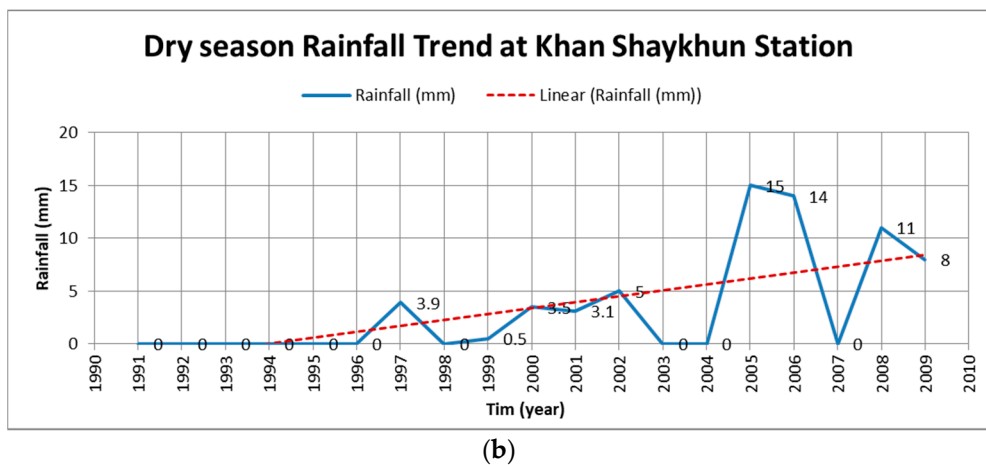

(**b**)

**Figure 6.** Rainfall trend at Khan Shaykhon station for the period (1991–2009). (**a**) Wet season.
(**b**) Dry season.

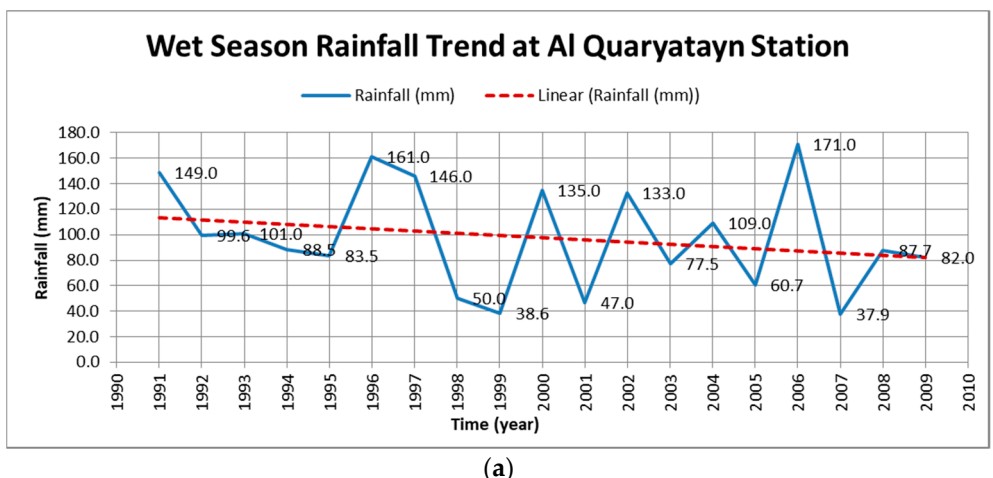

(**a**)

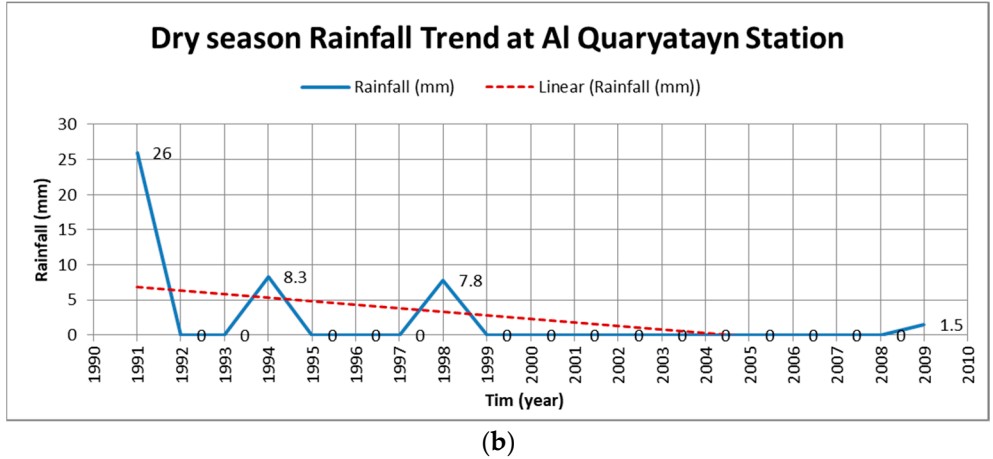

(**b**)

**Figure 7.** Rainfall trend at Al Quaryatayn station for the period (1991–2009). (**a**) Wet season.
(**b**) Dry season.

Figure 8 shows the analysis of rainfall trends at Tal Kalakh station. The highest rainfall
rate was recorded in 2002 (1552 mm) and the lowest rainfall rate was recorded in 1998
(427 mm) in the wet season. On the other hand, in the dry season the highest rainfall rate
was recorded in 2009 (104 mm), and the lowest rainfall rate was (0 mm) in 1993, 1995,
2003, 2004, and 2007. The rainfall has an increasing trend in both wet and dry seasons
(Figure 8a,b).

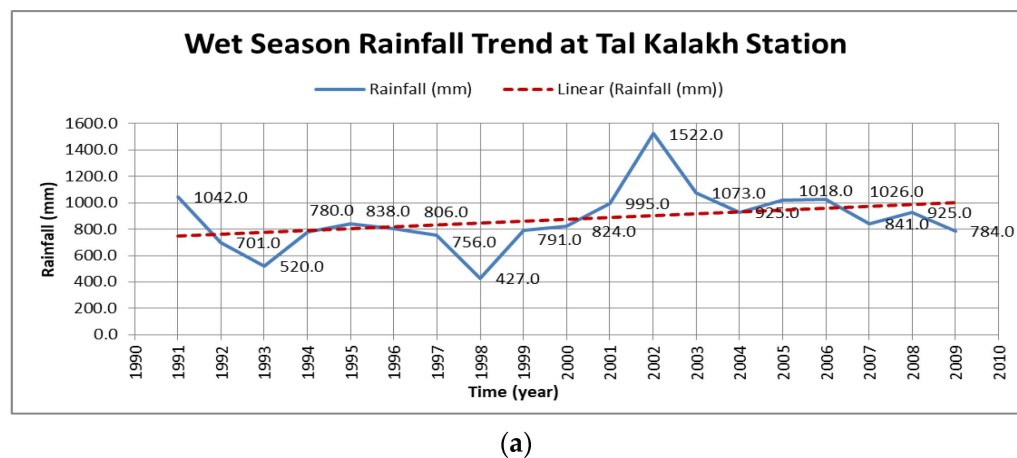

(**a**)

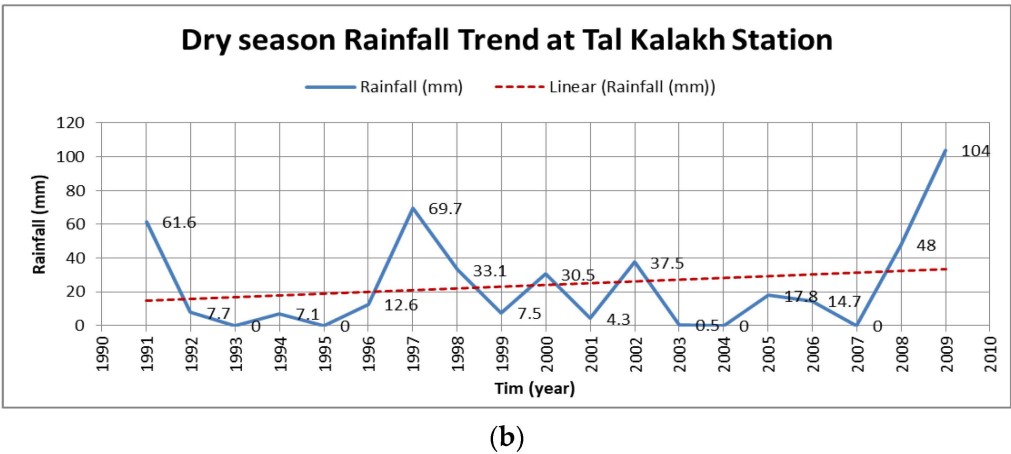

(**b**)

**Figure 8.** Rainfall trend at Tal Kalakh station for the period (1991–2009). (**a**) Wet season. (**b**) Dry season.

For Al Nabak station, the analysis of rainfall trends is shown in Figure 9. The highest rainfall rate in the wet season was recorded in 1991 (353 mm) and the lowest rainfall rate was recorded in 1999 (48.2 mm). The rainfall has a decreasing trend in the wet season (Figure 9a). In the dry season, the highest rainfall rate was recorded in 2004 (45.8 mm) and the lowest rate was (0 mm) in 1996, 2001, and 2003. The rainfall has a nearly constant trend in the dry season (Figure 9b).

At Al Busayrah station, the highest rainfall rate in the wet season was recorded in 2005 (223 mm) and the lowest rate was recorded in 2007 (31 mm). The rainfall has an increasing trend in the wet season (Figure 10a). In the dry season, the highest rainfall rate was recorded in 1994 (1.6 mm) and the lowest rate was (0 mm) in other years. The trend of rainfall is almost constant in the dry season (Figure 10b).

The analysis of rainfall trends for Al Tebni station is shown in Figure 11. In the wet season, the highest rainfall rate was recorded in 2000 (267 mm) and the lowest rate was recorded in 2007 (41.6 mm). The rainfall has a slight decreasing trend in the wet season (Figure 11a). However, in the dry season, the highest rainfall rate was recorded in 2005 (30 mm), and (0 mm) was recorded in most of the years except small rainfall in 1991, 1994, 2008, and 2009. The trend of rainfall has a slight increase in the dry season (Figure 11b).

For Ebla station, the analysis of rainfall trends is shown in Figure 12. The highest rainfall rate in the wet season was recorded in 2003 (710 mm) and the lowest rate was recorded in 1994 (374 mm). The rainfall provided a decreasing trend in the wet season (Figure 12a). In the dry season, the highest rainfall rate was recorded in 2009 (36.5 mm) but (0 mm) was recorded in 1991, 1993, 1995, 2004, and 2007. The rainfall has an increasing trend in the dry season (Figure 12b).

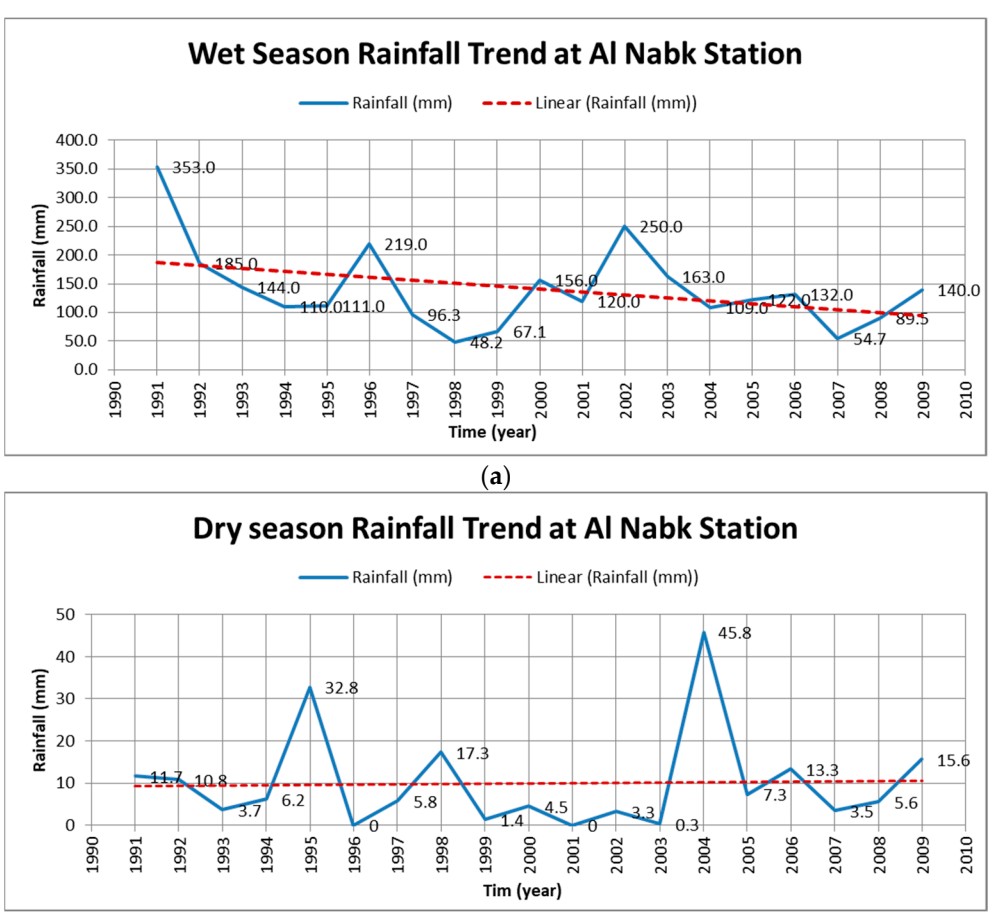

**Figure 9.** Rainfall trend at Al Nabak station for the period (1991–2009). (**a**) Wet season. (**b**) Dry season.

At Al Qadmus station, the highest rainfall rate in the wet season was recorded in 2002 (1634 mm) and the lowest rate was recorded in 2007 (868 mm). The rainfall has a decreasing trend in the wet season (Figure 13a). However, in the dry season the highest rainfall rate was recorded in 2009 (173 mm) and (1.0 mm) in 1991 and 2003. The trend of rainfall is increasing in the dry season (Figure 13b).

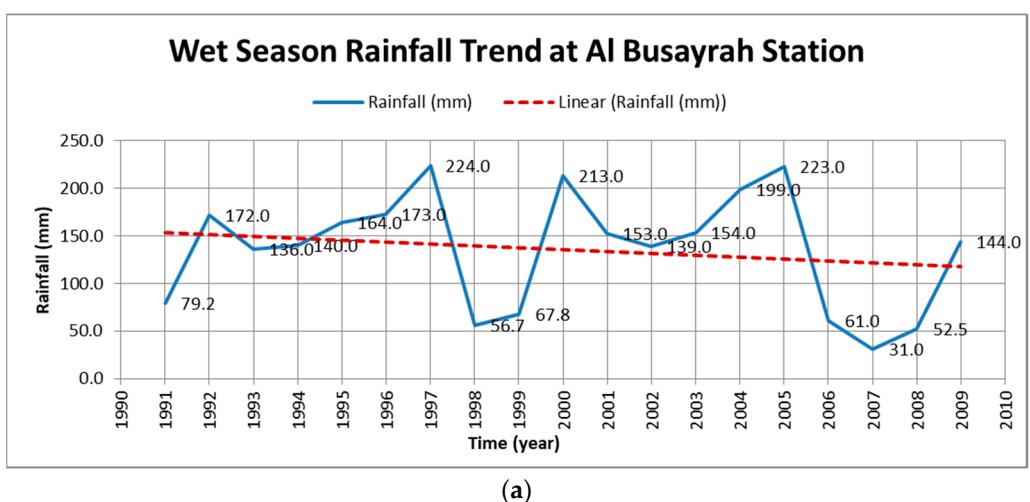

**Figure 10.** *Cont.*

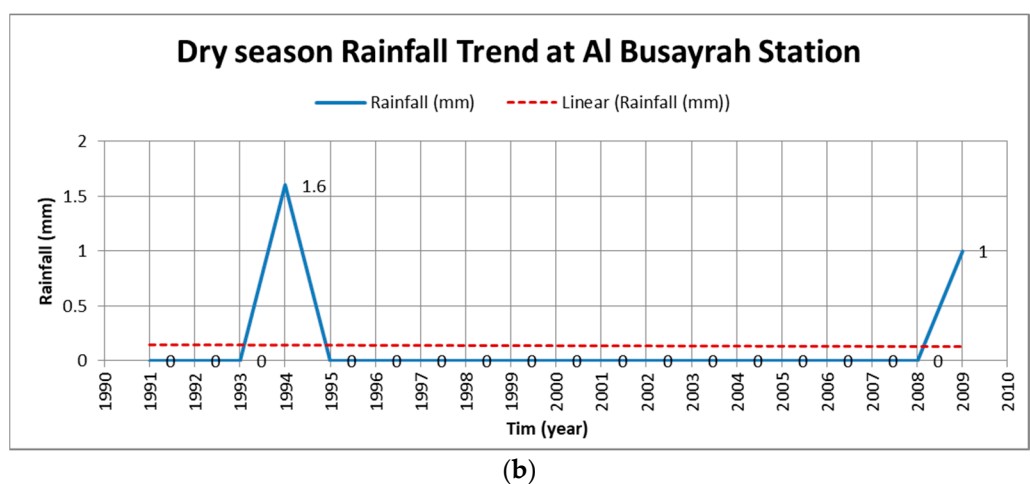

(**b**)

**Figure 10.** Rainfall trend at Al Busayrah station for the period (1991–2009). (**a**) Wet season. (**b**) Dry season.

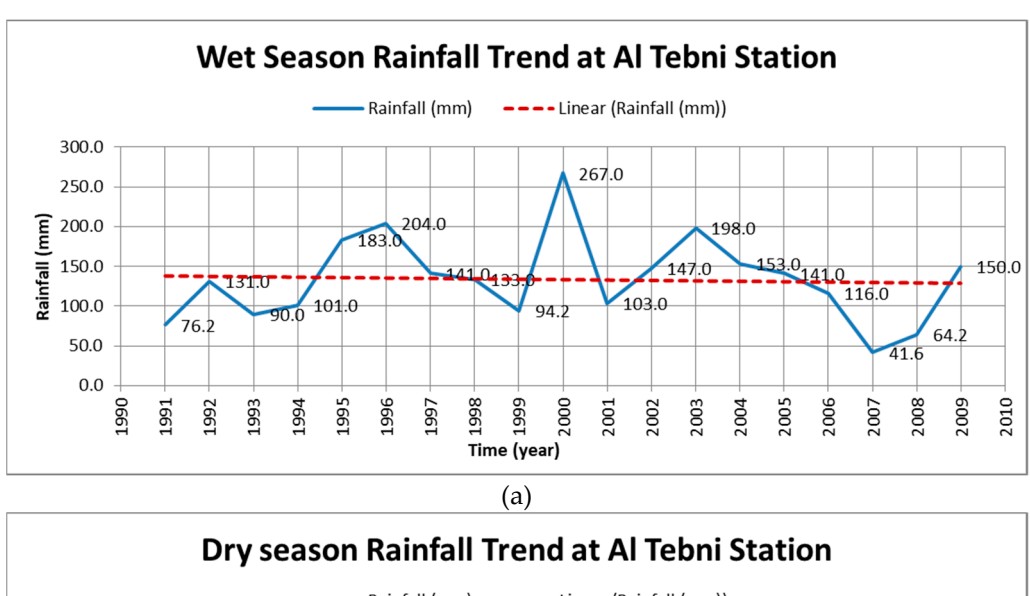

(a)

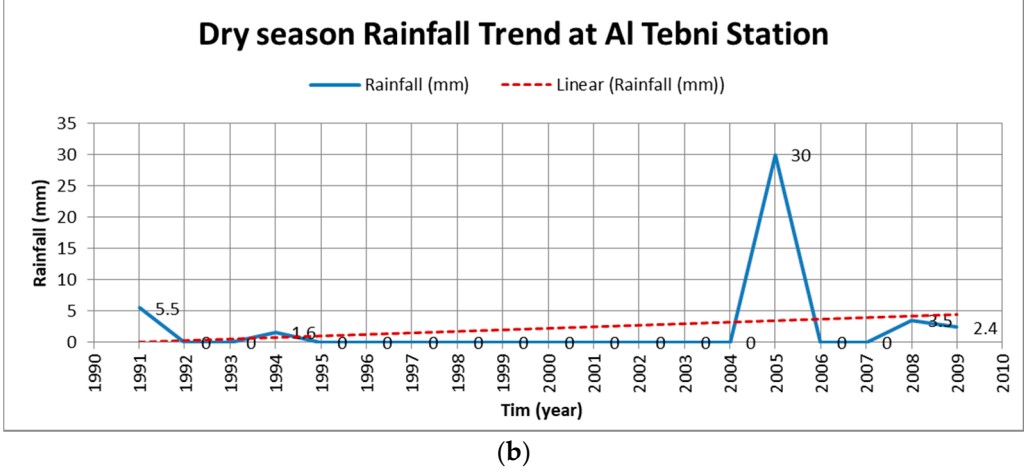

(**b**)

**Figure 11.** Rainfall trend at Al Tebni station for the period (1991–2009). (**a**) Wet season. (**b**) Dry season.

The analysis of rainfall trends at Baniyas is shown in Figure 14. In the wet season, the highest rainfall rate was recorded in 2002 (1216 mm) and the lowest rate was recorded in 2000 (474 mm). The trend of rainfall is decreasing in the wet season (Figure 14a). However, in the dry season, the highest rainfall rate was recorded in 1996 (194 mm), and (0 mm) was recorded in 1992 and 1993. The trend of rainfall is increasing in the dry season (Figure 14b).

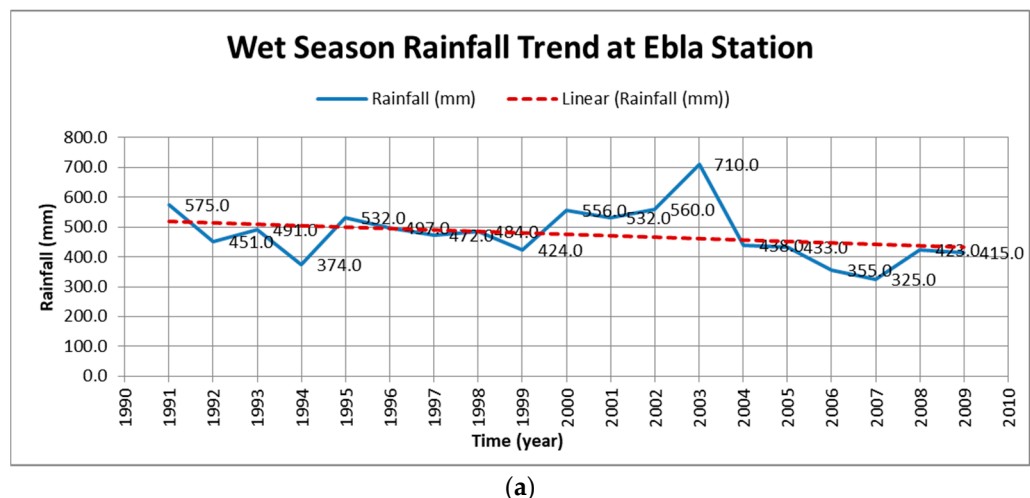

(**a**)

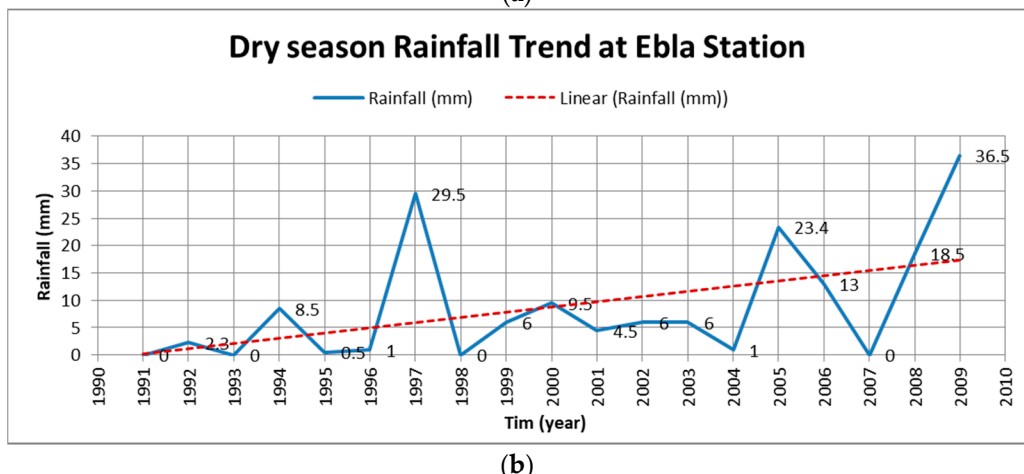

(**b**)

**Figure 12.** Rainfall trend at Ebla station for the period (1991–2009). (**a**) Wet season. (**b**) Dry season.

For Ar Reqqah station, the analysis of rainfall trends is shown in Figure 15. The highest rainfall rate in the wet season was recorded in 1996 (243 mm) and the lowest rate was recorded in 1999 (80.8 mm). The results show decreasing trend in the wet season (Figure 15a). In the dry season, the highest rainfall rate was recorded in 2009 (11 mm) but (0 mm) was recorded in most of the years except small values in 1994, 1997, and 2003. The rainfall has increasing trend in the dry season (Figure 15b).

Figure 16 shows the analysis of rainfall trends at Ain Issa station. In the wet season, the highest rainfall rate was recorded in 1995 (337 mm) and the lowest was recorded in 2007 (73 mm). The rainfall has a decreasing trend in the wet season (Figure 16a). However, in the dry season, the highest rainfall rate was recorded in 2009 (10.3 mm), and (0 mm) was recorded for the other years except 1994 (8.0 mm). The trend of rainfall is increasing in the dry season (Figure 16b).

Figures 6–16 presented the rainfall trends in Syria for the period (1991–2009) at the significant stations in the wet and dry seasons. It is noticed that most stations have constant or slight increasing rainfall trends in the dry season. Significant increasing trends were proven at stations: Tal Kalakh (Figure 8), Al Tebni (Figure 11), Al Qadmus (Figure 13), and Baniyas (Figure 14), located in the southwestern part of the Syria on the Mediterranean Sea. However, decreasing trends in the evaluated period (19 years) has been noticed in the wet season at all stations except one station (Tal Kalakh), located at a high level on the Mediterranean Sea shore. The significant negative trends in rainfall time series were shown in stations Ar Reqqah (Figure 15) and Ain Issa (Figure 16). The stations Ebla (Figure 12), Khan Shaykhun (Figure 6), Al Quaryatayn (Figure 7), and Al Busayrah (Figure 10) proved significant decreasing trends in rainfall. These stations (except Al Quaryatayn) are situated

in the northwestern part of Syria. The wet season is the most important one as the country receives more rain in this period. The decreasing rainfall trend at most of the stations in this period means shortage of water and drought in the country.

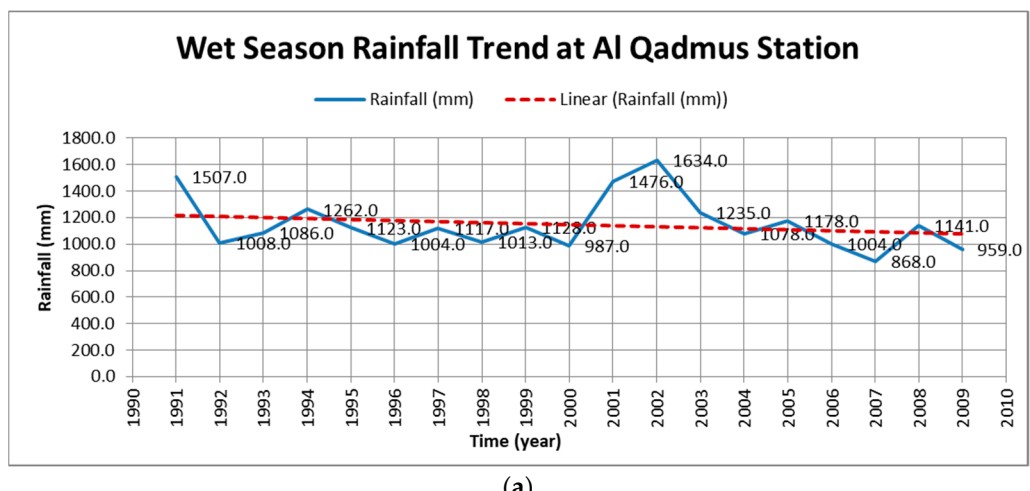

(a)

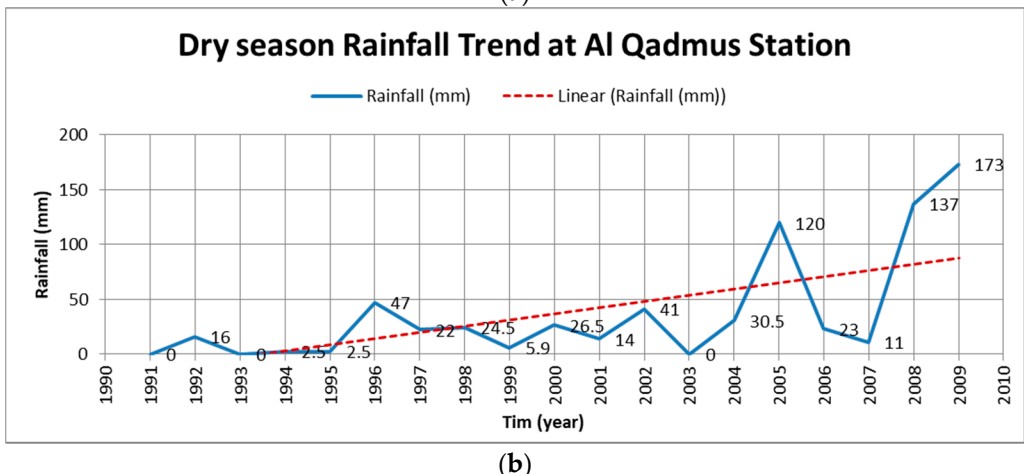

(b)

**Figure 13.** Rainfall trend at Al Qadmus station for the period (1991–2009). (**a**) Wet season. (**b**) Dry season.

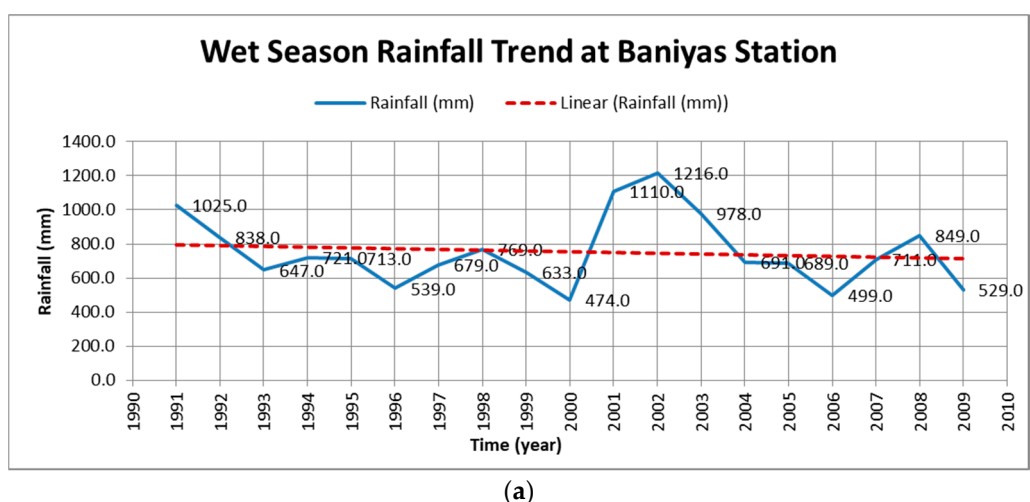

(a)

**Figure 14.** *Cont.*

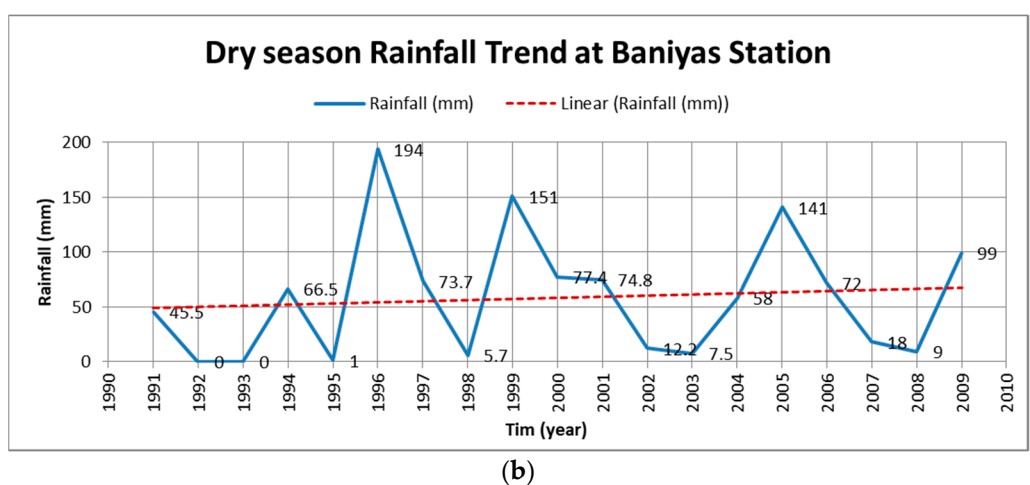

(**b**)

**Figure 14.** Rainfall trend at Baniyas Station for the period (1991–2009). (**a**) Wet season. (**b**) Dry season.

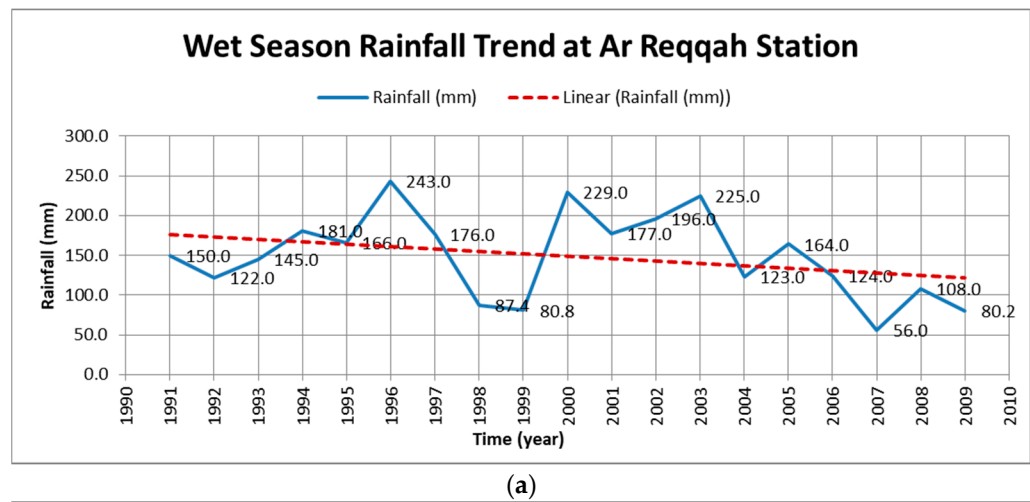

(**a**)

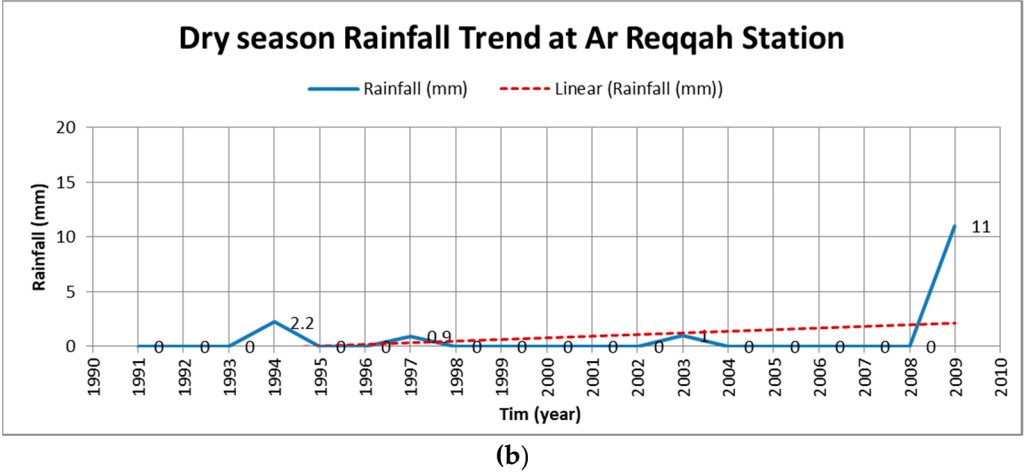

(**b**)

**Figure 15.** Rainfall trend at Ar Reqqah station for the period (1991–2009). (**a**) Wet season. (**b**) Dry season.

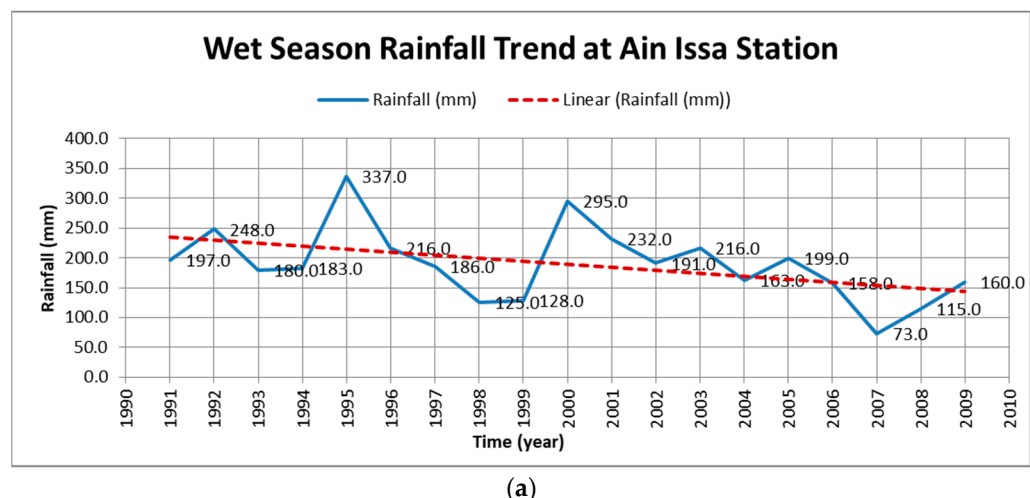

(**a**)

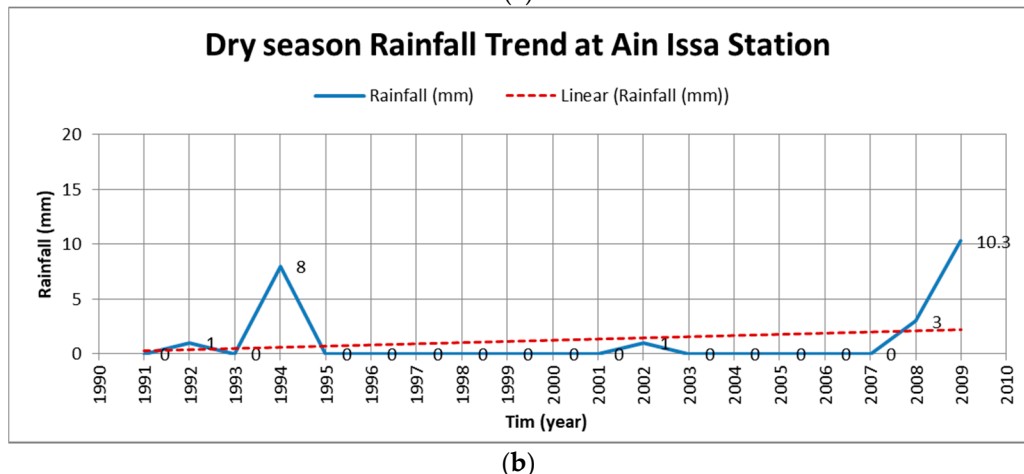

(**b**)

**Figure 16.** Rainfall trend at Ain Issa station for the period (1991–2009). (**a**) Wet season. (**b**) Dry season.

*4.2. Mann–Kendal Results*

The most frequent statistical approach for analyzing time series datasets is the non-parametric Mann–Kendall trend test. The rainfall at 71 stations in Syria was analyzed using Mann–Kendall trend for the period (1991–2009). The following sections present the results of Mann–Kendall trends for monthly and long period.

4.2.1. Result of Rainfall Analysis by Mann–Kendall for Monthly Trends

In this section, the trend analysis is focused on the monthly trends from September to May because there is no rainfall during June, July, and August. Mann–Kendall analysis is used for monthly trends. The results achieved for the period (1991–2009) at all stations are analyzed. Figure 17 shows the trends calculated by Mann–Kendall in September, October, and November. The rainfall amount is very small in September and the trend is increasing at all stations (Figure 17a). Additionally, the trend is increasing in October at most of the stations (Figure 17b), but in November the trend is decreasing at most of the stations (Figure 17c).

The rainfall trends by Mann–Kendall for months December, January, and February are shown in Figure 18. The trend in December and January is decreasing in most of the stations (Figure 18a,b). However, trend is increasing in February in a large number of the stations (Figure 18c).

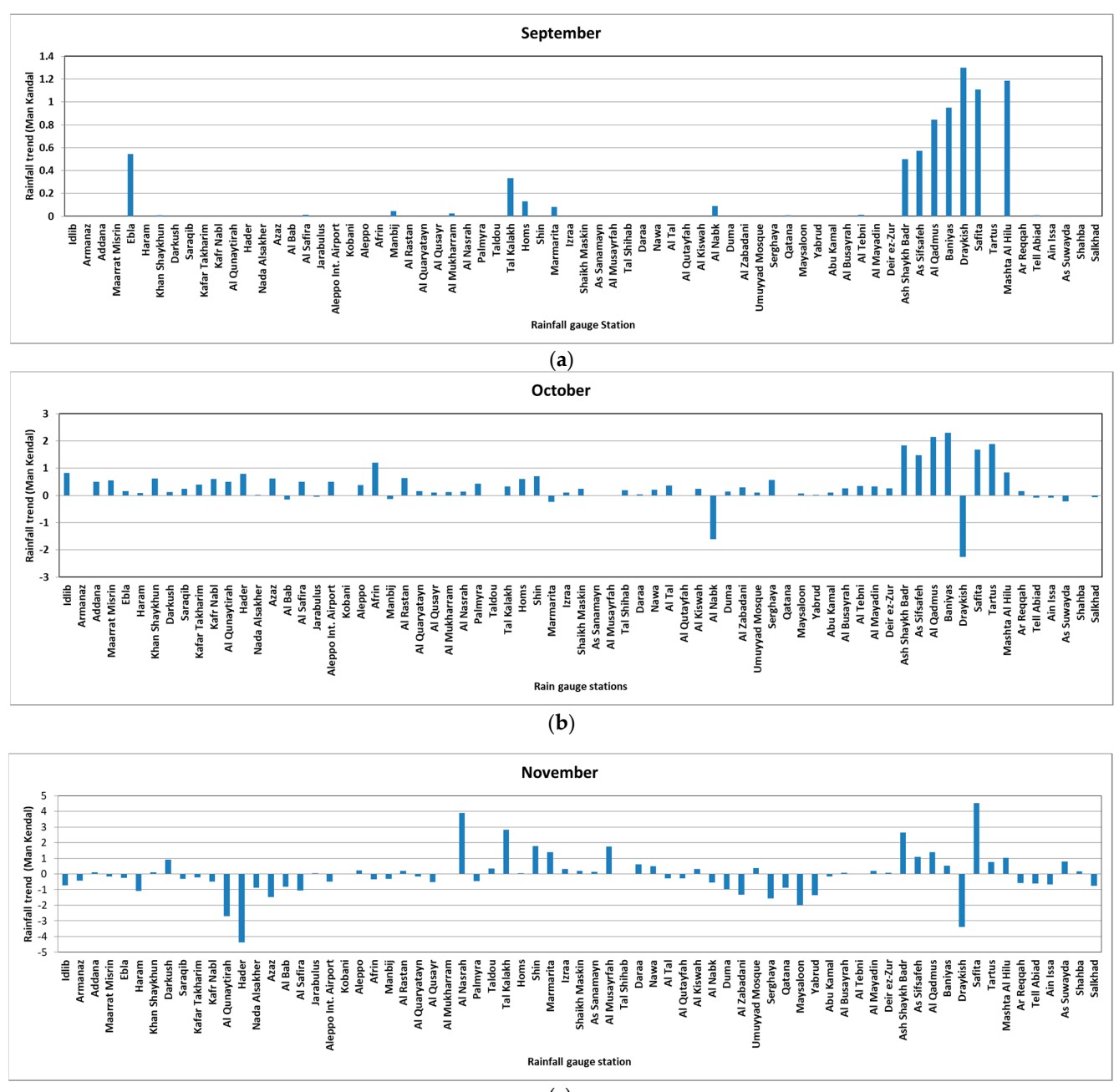

**Figure 17.** Rainfall trends (Mann–Kendall) for months (September–November) for the period (1991–2009). (**a**) September, (**b**) October, (**c**) November.

The Mann–Kendall trend is shown in Figure 19 for months March, April, and May. The trend is decreasing in March and May almost at all stations (Figure 19a,c) and increasing in April at most of the stations (Figure 19b).

The analysis of the monthly rainfall data presented in Figures 17–19 showed that there was a decrease in the rainfall trend in May and an increasing trend in September. Statistically significant negative rainfall trends were identified at the Qatana and Maysaloon stations which are situated in mountainous area near the capital city (Damascus) in the north of Syria. The highest decrease in rainfall, up to 4.67 mm/month (December), which corresponds to 5% of December's rainfall during the evaluated period, occurs at Maysaloon station. An increase in rainfall up to 3 mm/month (April), which corresponds to 10% of April's rainfall during the evaluated period, was found at the Tal Kalakh and Shin stations which are situated near the Mediterranean Sea.

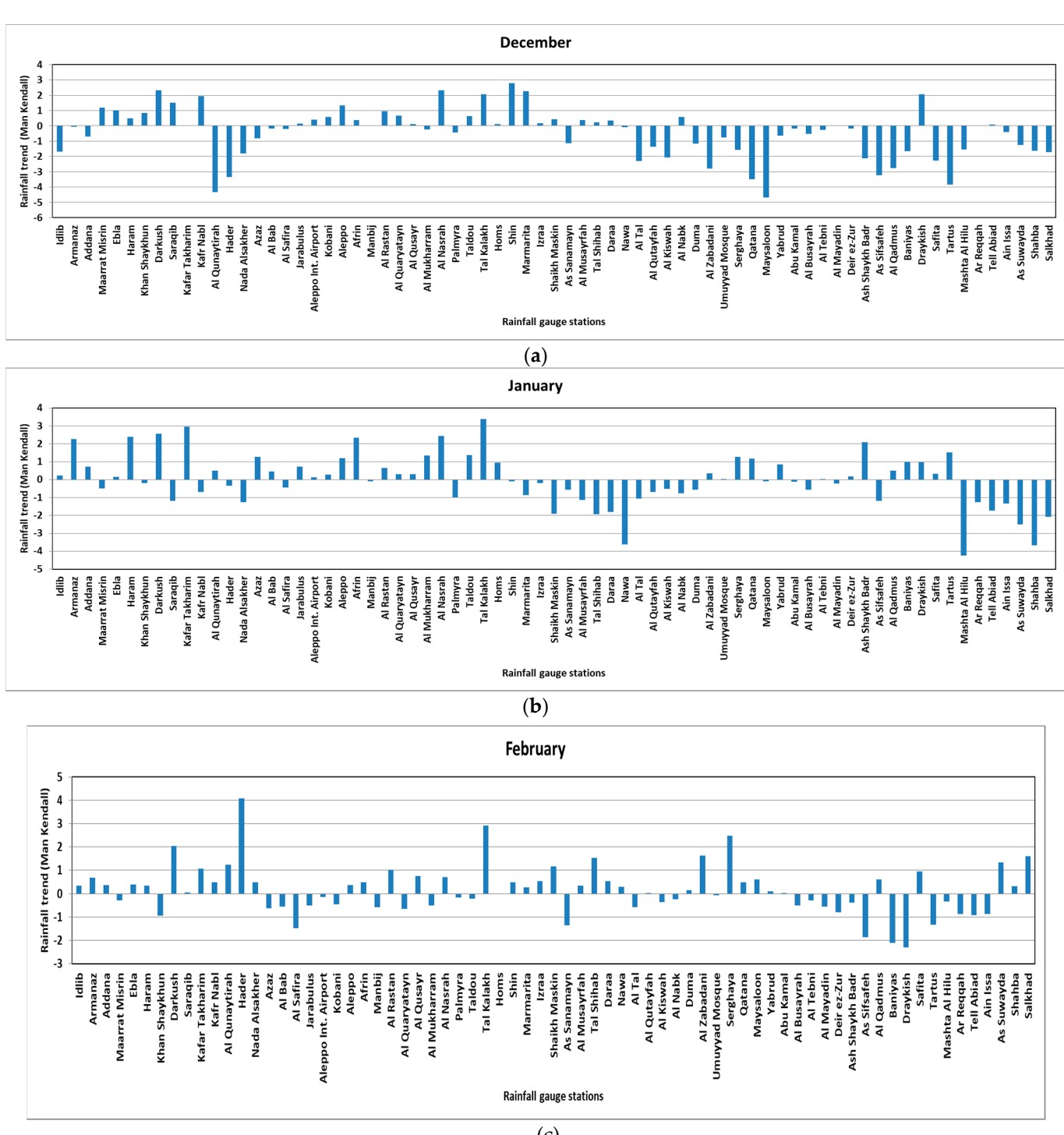

**Figure 18.** Rainfall trends (Mann–Kendall) for months (September–November) for the period (1991–2009). (**a**) December, (**b**) January, (**c**) February.

4.2.2. Results of Rainfall Analysis by Mann–Kendall for Long Periods

Results of seasonal rainfall trends are presented in Figure 20. Monthly data series for 19 years (1991–2009), were considered for trend detection. The evaluation was conducted for the period from September to May, as rainfall in other months was almost zero. The magnitude of the trend is expressed by Sen's estimator of the slope of all the data points.

The results of the long-term trend analysis are shown in Figure 20. The months from September to November showed increasing rainfall trends (Figure 20a), despite the fact that this is a dry season. In the 19-year period studied, the winter season from December

to February revealed decreasing trends in rainfall (Figure 20b). During the months from March to May, decreasing trends were observed at all stations (Figure 20c). The results from Figure 20 match with the results of monthly trend analysis as shown in Figures 17–19 and the trend analysis at significant stations Figures 6–16.

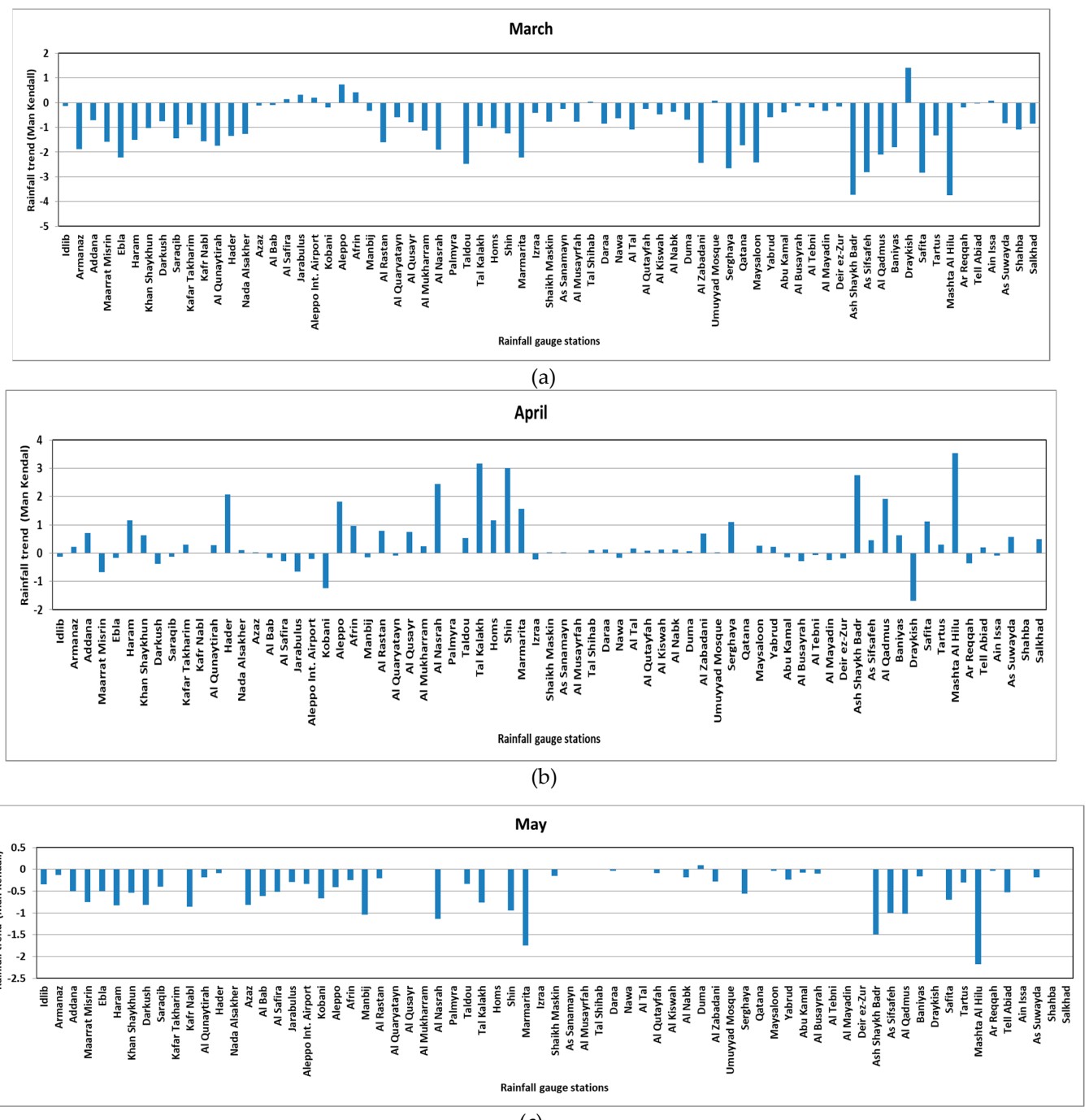

**Figure 19.** Rainfall trends (Mann–Kendall) for months (March–May) for the period (1991–2009). (**a**) March, (**b**) April, (**c**) May.

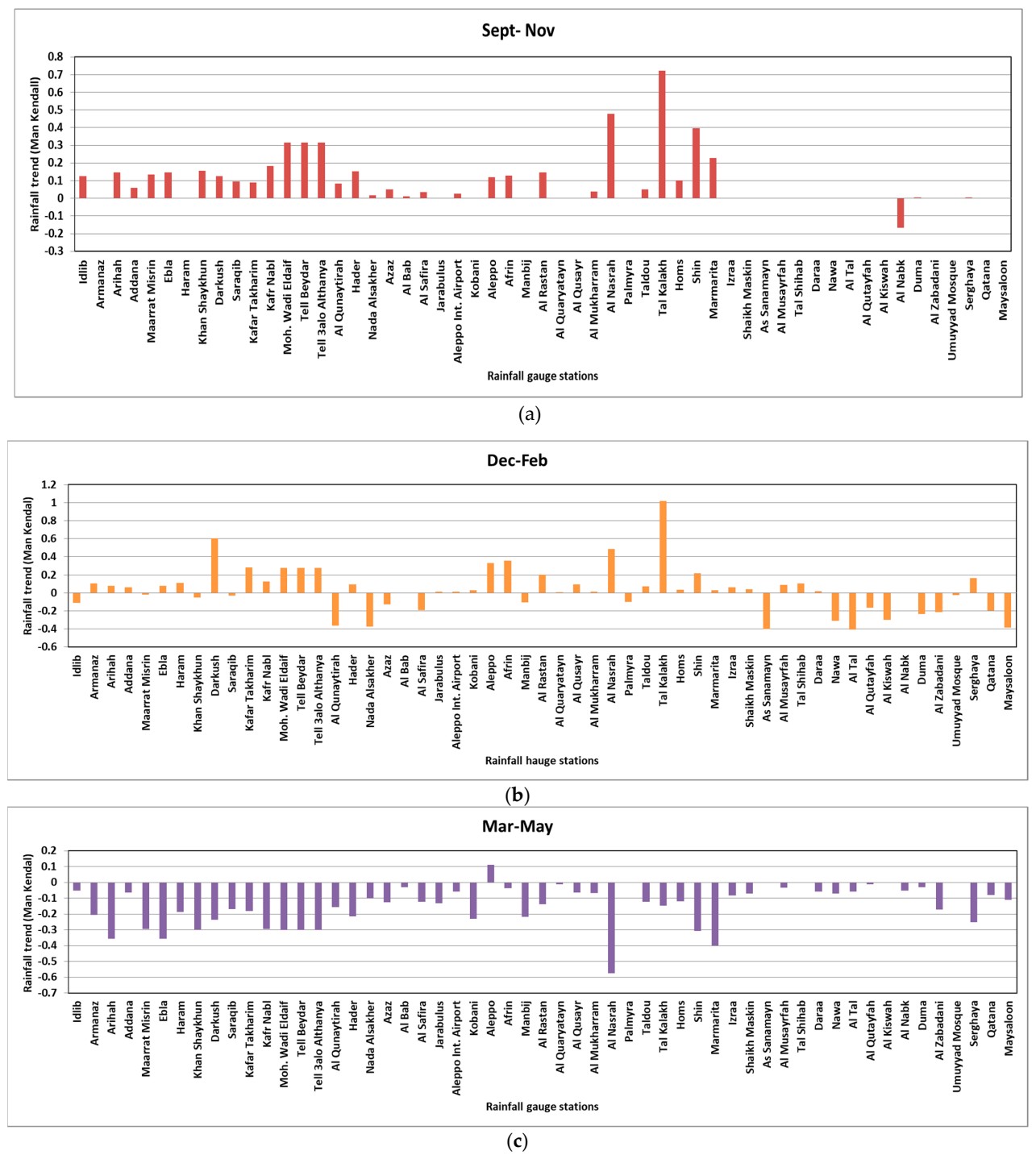

**Figure 20.** Rainfall trends (Mann–Kendall) at all stations (71) for the period (1991–2009). (**a**) September–November, (**b**) December–February, (**c**) March–May.

*4.3. Stations of Significant Trends*

Results of rainfall trend analysis and seasonal trends are presented in Table 1 (significant trends that were occurring during the evaluated period in evaluated rain gauge stations). Monthly data series for the 19 years' period, from 1991–2009, were considered for trend detection.

Comparison between the significant stations is presented in Figure 21. The trend analysis in the three periods (September to November), (December to February), and (March to May) are presented in Figure 21a–c, respectively. The results proved that the rainfall

trend has increased at most of the stations in the period from September to November and decreased in the periods December to February and March to May.

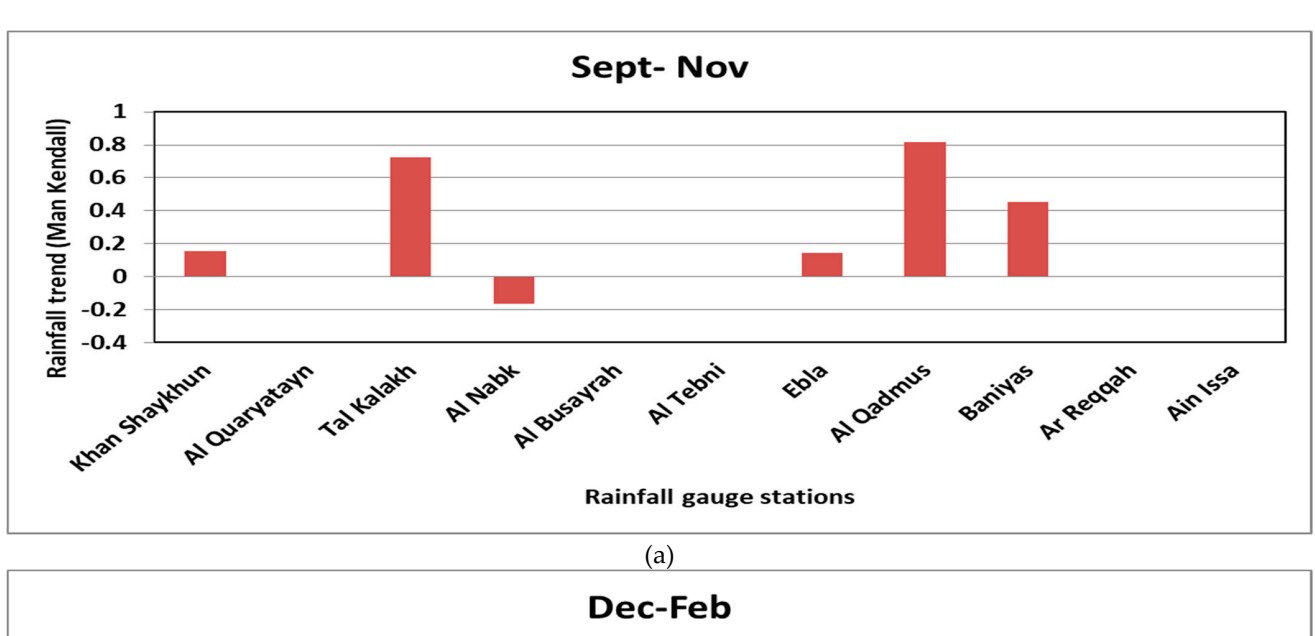

(a)

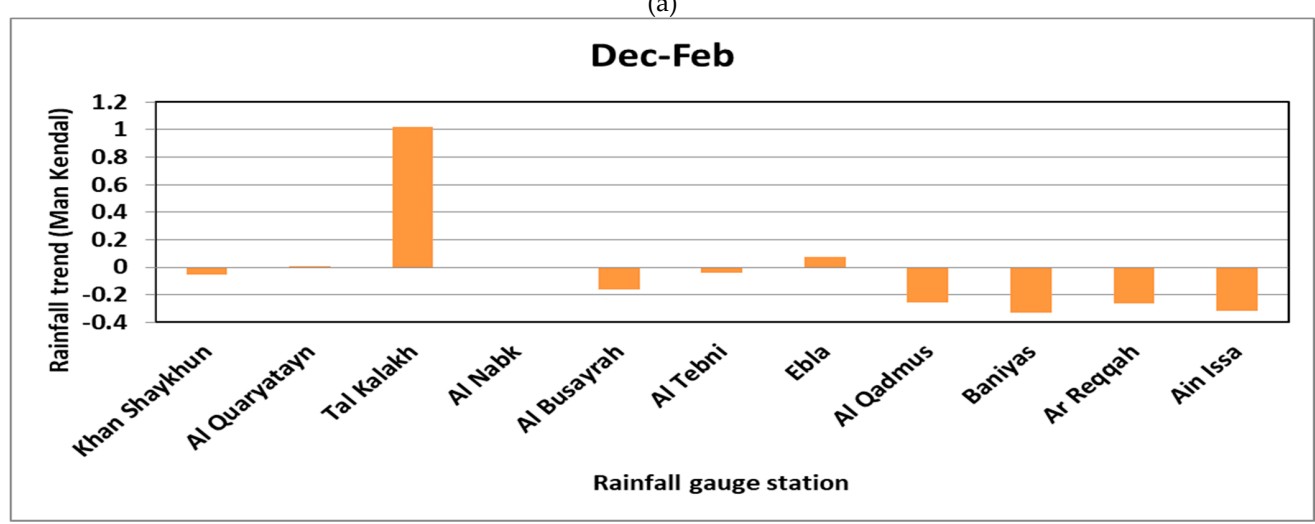

(b)

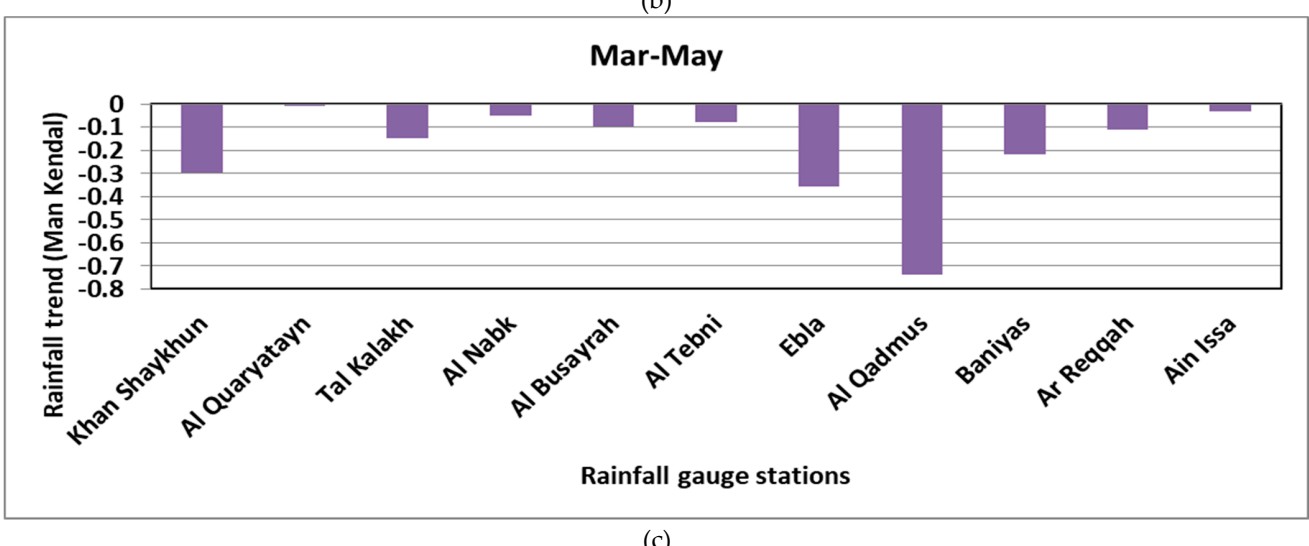

(c)

**Figure 21.** Rainfall trends (Mann–Kendall) for significant trend stations in the period (1991–2009). (**a**) September–November (**b**) December–February (**c**) March–May.

From the evaluation, it is obvious that slight increasing trends in rainfall in Syria are occurring in the fall period from September to November. However, from December to May, i.e., winter and spring periods, significant decreasing trends are occurring.

### 4.4. Evaluation of Rainfall Trends in Syria

In this section, ArcGIS (Geostatistical Analyst) modeling and analysis tools are utilized to model the geographical distribution of rainfall in Syria. Geostatistics is a branch of statistics that deals with the regionalization of random variables in a specific area. A random function was generated by a set of random variables. The random function model is based on the experimental variogram, which is a study of the spatial variability of the researched phenomenon in different directions. This research yielded a mathematical model of variogram defined by anisotropy and autocorrelation in altering spatial variability in distinct directions of space. The formula for calculating the empirical semi-variogram is written as the following [45]:

$$\gamma(h) = \frac{1}{2n(h)} \sum_{i=1}^{n(h)} [z(s_i) - z(s_i + h)]^2 \tag{6}$$

where $\gamma(h)$ is the estimated semi-variation for the distance $h$; $n(h)$ is the number of pairs of measured points separated by a distance $h$; and $z(s_i)$ is the measured value in point $(s_i)$.

The empirical semi-variogram was calculated first. The empirical semi-variogram was then transferred to the theoretical model and its parameters are determined. The experimental and theoretical assumptions that go into developing a model for a particular set of data are crucial. The real technique of estimating the phenomenon of unknown values based on known data, Kriging [42], follows the determination of semi-variogram parameters. Figure 22a–c show the spatial distribution of trend analysis for rainfall time series. The figure shows how the seasonal distribution of rainfall over Syria, determined by data from 71 climatic stations, varied substantially from place to place and over time. The blue color represents increasing trends in rainfall and the red color represents decreasing trends in rainfall.

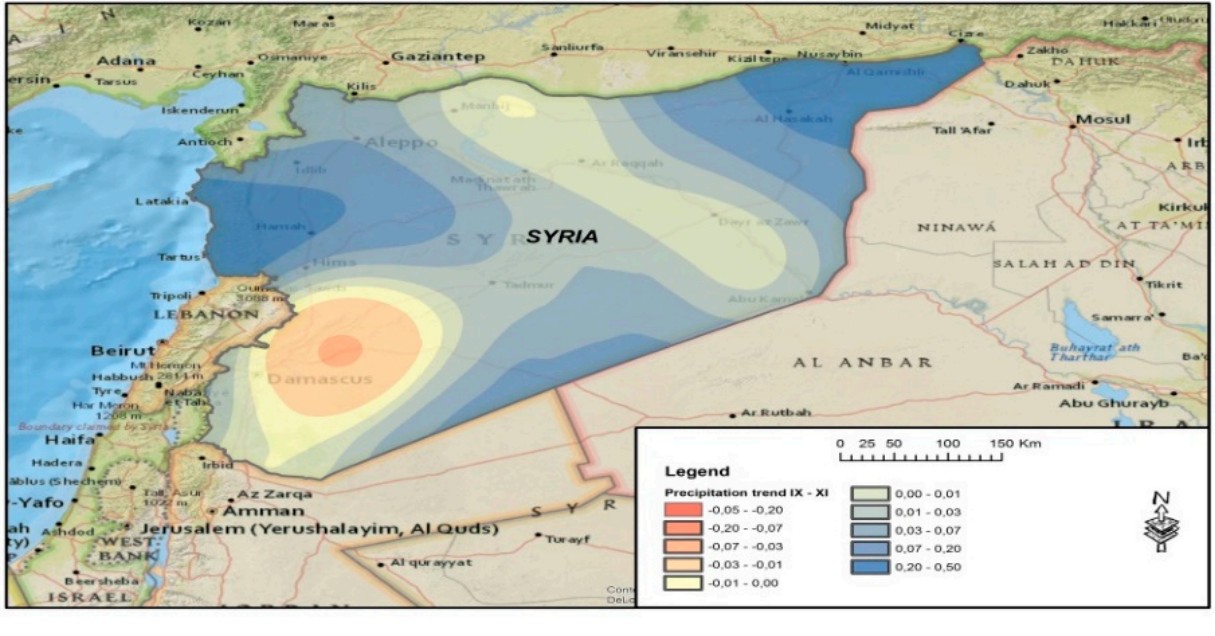

(**a**)

**Figure 22.** *Cont.*

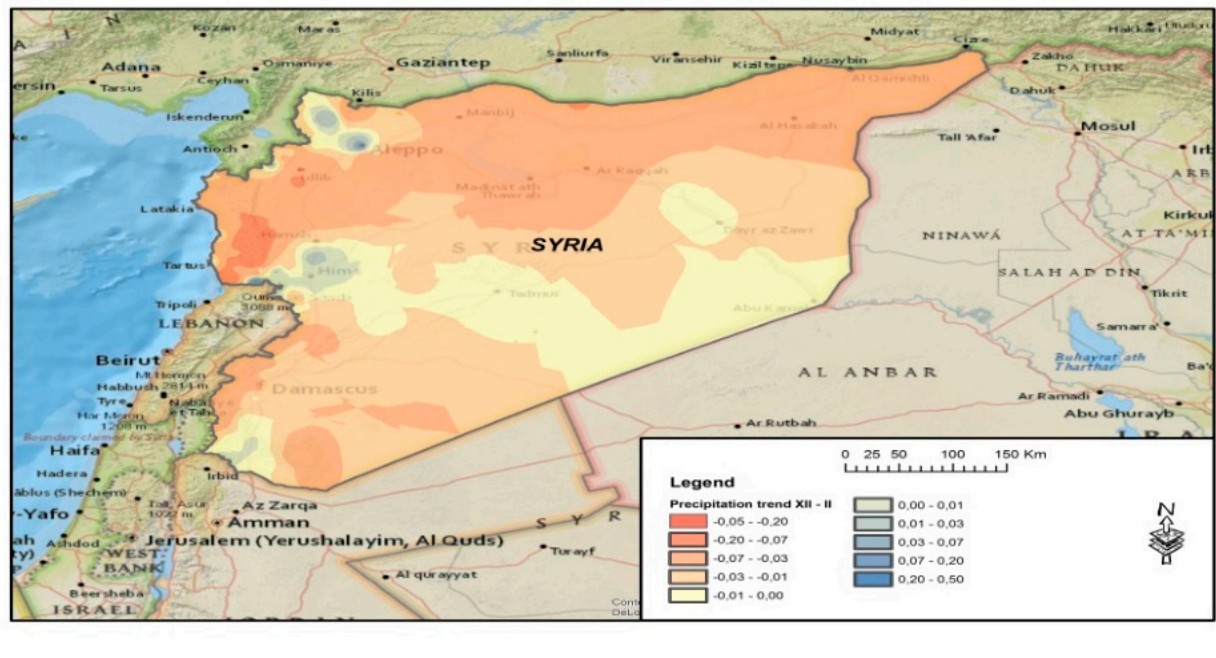

(**b**)

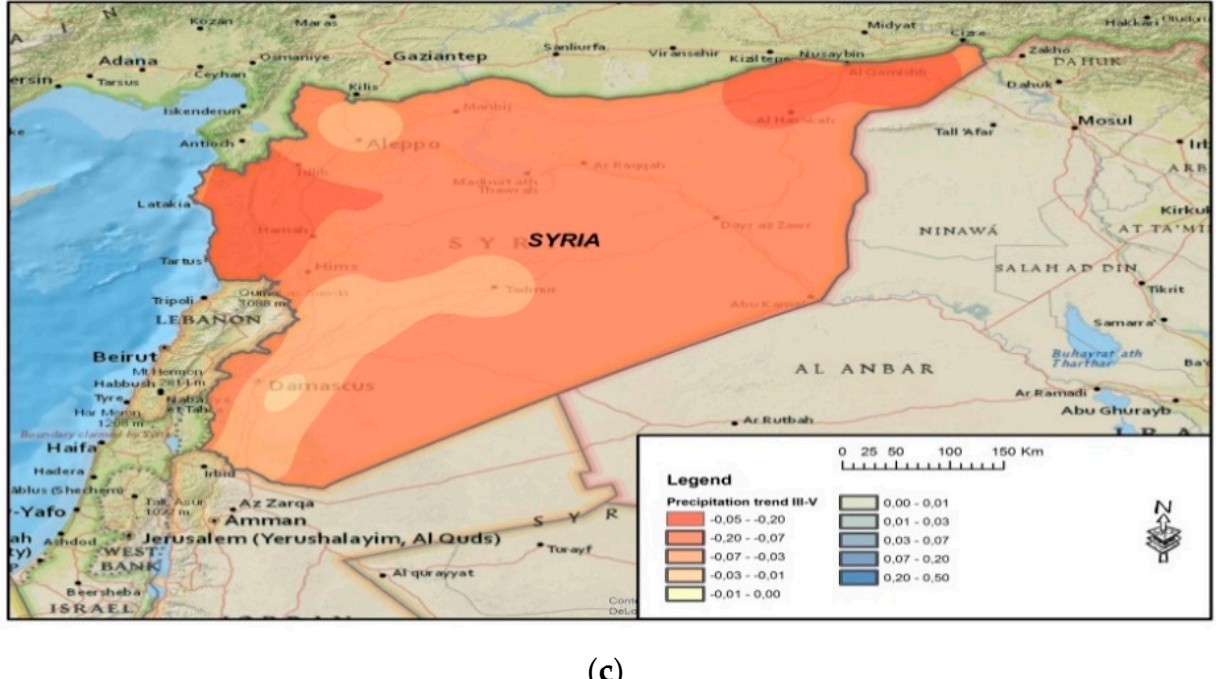

(**c**)

**Figure 22.** Spatial distribution of rainfall trends in Syria. (**a**) September to November. (**b**) December to February. (**c**) March to May.

According to the fifth report released by (IPCC, 2014) [46], over the period (1900–2005), the precipitation has decreased across the Mediterranean and southern Asia and increased over latitudes north of 30. The current results of spatial and temporal distribution of rainfall over Syria as a Mediterranean country is consistent with the decrease in rainfall that the IPCC reports. This reveals that Syria will be affected by climate change, and appropriate measured must be considered to adapt with such changes.

## 5. Conclusions

Climate change is likely to have continuous impact on the hydrological cycle parameters. These impacts could be carefully considered in water resources management. Significant natural factors related to climate change are emphasized in this study, including temperature, wind, and precipitation which will affect water resources at various levels. Syria is considered a water scarce country, and population growth and industrial development will increase water needs. On the other hand, climate change could have a clear impact on water resources in Syria. This paper shows how climate change could affect the rainfall trends. The special and temporal variability of rainfall trends in response to climate change in Syria in the period (1991–2009) were presented and discussed. Data were collected from 71 stations distributed all over the country for rainfall trends analysis. The Mann–Kendall test was used for rainfall trends analysis in monthly and seasonal scales, and the results obtained indicated significant decreasing trends, consistent with the fifth report by the IPCC. Additionally, rainfall trend analysis was conducted for wet and dry seasons, and the results showed that most stations have constant or slightly increasing rainfall trends in the dry season. However, decreasing trends in the evaluated period (19 years) has been noticed in the wet season at almost all stations. The wet season is the most important one as the country receives more rain in this period. Decreasing the rainfall trend at most of the stations in this period means shortage of water and drought in the country. As most rainfall occurs in the winter and spring, infrequently in the fall, and rarely in the summer, the country will face water shortages in the future due to expected climate change. The outcomes of this study could help in manufacturing future plans of water resource management in Syria.

**Author Contributions:** Conceptualization, M.Z.; methodology, M.Z., H.F.A.-E., J.S. and I.A.; validation, K.K., P.P. and I.A.; formal analysis, M.Z., H.F.A.-E. and J.S.; investigation, H.F.A.-E. and I.A.; data curation, K.K., P.P. and I.A.; writing—original draft preparation, I.A., K.K. and H.F.A.-E.; writing—review and editing, M.Z., H.F.A.-E., J.S., K.K., P.P. and I.A.; supervision, M.Z. and H.F.A.-E.; project administration, K.K. and J.S.; funding acquisition, M.Z. All authors have read and agreed to the published version of the manuscript.

**Funding:** This research received no external funding.

**Institutional Review Board Statement:** Not applicable.

**Informed Consent Statement:** Not applicable.

**Data Availability Statement:** The data are not publicly available due to institutional property rights.

**Acknowledgments:** This work was supported by the Slovak Research and Development Agency under the contract no. APVV-17-0549. Thank you for the support of project KEGA 059TUKE-4/2019 M-learning tool for intelligent modeling of building site parameters in a mixed reality environment. This work was supported by the Slovak Research and Development Agency under the Contract no. APVV-20-0281. This work was supported by project of the Ministry of Education of the Slovak Republic VEGA 1/0308/20 Mitigation of hydrological hazards, floods and droughts by exploring extreme hydroclimatic phenomena in river basins.

**Conflicts of Interest:** The authors declare no conflict of interest.

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
