# Peer review of "Spatial and Temporal Variability of Rainfall Trends in Response to Climate Change—A Case Study: Syria"

_water, doi:10.3390/w14101670_

Round 1
Reviewer 1 Report
Title
The Title reflects the paper’s content accurately. Change the typo error “Special” to “Spatial”
Abstract
The Abstract determines the paper’s content and objectives in a very manifest and complete fashion.
- Introduction
In L38 add [1]. In L39-L42 change the order of sentences as human contribution is primary. In L43 there are no mitigating consequences stemming from global warming so delete it. Replace L48-L50 by “The major cause of global warming is the greenhouse gases. They include carbon dioxide, methane, nitrous oxides and in some cases chlorine and bromine containing compounds. The build-up of these gases in the atmosphere changes the radiative equilibrium in the atmosphere. Their overall effect is to warm the Earth’s surface and the lower atmosphere because greenhouse gases absorb some of the outgoing radiation of Earth and re-radiate it back towards the surface” (keeping the initials) [2] and its effects are seen in [3]. Add after that ‘as a result of climate change the occurrence of extreme weather events has increased [4]’. In L139 after ‘distribution’ the phrase ‘and the relationship is so close that and precipitation can be employed as an indicator of climate change [5]’. Otherwise, good work.
Transfer Methodology here.
- Study area and data preparation
Make this a part of ‘Results’
- Results
Separate Results from Discussion
- Conclusions
Precise and firmly based on the previous sections.
References
[1] Wolff, E., I. Fung, B. Hoskins, J. F. B. Mitchell, T. Palmer, B. Santer, J. Shepherd, K. Shine, S. Solomon, K. Trenberth, et al., “Climate Change: Evidence and Causes Update 2020,” Royal Society-US National Academy of Sciences, 2020.
[2] Abouelfadl, S., “Global Warming – Causes, Effects and Solution’S Trials,” JES. J. Eng. Sci., vol. 40, no. 4, pp. 1233–1254, 2012, doi: 10.21608/jesaun.2012.114490.
[3] Masson-Delmotte, V., P. Zhai, H.-O. Pörtner, D. Roberts, J. Skea, P. R. Shukla, A. Pirani, W. Moufouma-Okia, C. Péan, R. Pidcock, et al., Eds., Global warming of 1.5°C An IPCC Special Report on the impacts of global warming of 1.5°C above pre-industrial levels and related global greenhouse gas emission pathways, in the context of strengthening the global response to the threat of climate change. Geneva, Switzerland: Intergovernmental Panel on Climate Change., 2018.
[4] Attribution of Extreme Weather Events in the Context of Climate Change. National Academies Press., 2016. doi: 10.17226/21852.
[5] Panagoulia, D., “Assessment of daily catchment precipitation in mountainous regions for climate change interpretation,” Hydrol. Sci. J., vol. 40, no. 3, pp. 331–350, 1995.
Author Response
We would like to thank the reviewer for valuable comments that have certainly improved the quality of the paper. Please find enclosed the revised paper in which the and reviewer comments have been addressed. The reviewer's comments have been addressed in the same order as in the reviewer's report.
Title
The Title reflects the paper’s content accurately. Change the typo error “Special” to “Spatial”
Response: Thanks, the word has been changed to Spatial.
Abstract
The abstract determines the paper’s content and objectives in a very manifest and complete fashion.
Response: Thanks to the reviewer for his positive comments.
Introduction
In L38 add [1].
Response: It has been added to line 39.
In L39-L42 change the order of sentences as human contribution is primary.
Response: The sentence has been changed in line 42.
In L43 there are no mitigating consequences stemming from global warming so delete it.
Response: It has been changed in line 43.
Replace L48-L50 by “The major cause of global warming is the greenhouse gases. They include carbon dioxide, methane, nitrous oxides and in some cases chlorine and bromine containing compounds. The build-up of these gases in the atmosphere changes the radiative equilibrium in the atmosphere. Their overall effect is to warm the Earth’s surface and the lower atmosphere because greenhouse gases absorb some of the outgoing radiation of Earth and re-radiate it back towards the surface” (keeping the initials) [2] and its effects are seen in [3].
Response: It has been replaced in lines 47-52.
Add after that ‘as a result of climate change the occurrence of extreme weather events has increased [4]’.
Response: It has been replaced in line 52-53.
In L139 after ‘distribution’ the phrase ‘and the relationship is so close that and precipitation can be employed as an indicator of climate change [5]’.
Response: It has been changed in line 172.
Otherwise, good work
Response: Thanks to the reviewer for his positive comments.
Study area and data preparation
Make this a part of ‘Results’
Response: Thanks, but based on the editor and other reviewers comments it is better to keep it as separate part.
Results
Separate Results from Discussion
Response: Results and discussion put in one part because the results are divided into 4 parts and each part includes the results and discussion of results. Separation may cause some repetition.
Conclusions
Precise and firmly based on the previous sections.
Response: Thanks to the reviewer for his positive comments.
References
- Wolff, E., I. Fung, B. Hoskins, J. F. B. Mitchell, T. Palmer, B. Santer, J. Shepherd, K. Shine, S. Solomon, K. Trenberth, et al., “Climate Change: Evidence and Causes Update 2020,” Royal Society-US National Academy of Sciences, 2020.
- Abouelfadl, S., “Global Warming – Causes, Effects and Solution’S Trials,” JES. J. Eng. Sci., vol. 40, no. 4, pp. 1233–1254, 2012, doi: 10.21608/jesaun.2012.114490.
- Masson-Delmotte, V., P. Zhai, H.-O. Pörtner, D. Roberts, J. Skea, P. R. Shukla, A. Pirani, W. Moufouma-Okia, C. Péan, R. Pidcock, et al., Eds., Global warming of 1.5°C An IPCC Special Report on the impacts of global warming of 1.5°C above pre-industrial levels and related global greenhouse gas emission pathways, in the context of strengthening the global response to the threat of climate change. Geneva, Switzerland: Intergovernmental Panel on Climate Change., 2018.
- Attribution of Extreme Weather Events in the Context of Climate Change. National Academies Press., 2016. doi: 10.17226/21852.
- Panagoulia, D., “Assessment of daily catchment precipitation in mountainous regions for climate change interpretation,” Hydrol. Sci. J., vol. 40, no. 3, pp. 331–350, 1995.
Response: Thanks, all of these references have been added to the paper and list of references.
Reviewer 2 Report
The paper “Special and temporal variability of rainfall trends in response to climate change-A case Study: Syria” is addressing timely important aspects but the paper needs to improve by addressing the bellow detail comments before publishing the paper.
Abstract
It is very important to provide the quantitative observation of the study in the Abstract to strengthening it.
Introduction
- References are missing in Lines 51-52, 62-65, 66-68, 4-88,
- The introduction is too lengthy, and it is important to reduce the length by highlighting important points related to the topic.
- It is important to provide the recent and similar studies in the introduction on the rainfall trend analysis in different geographical context of the world as some of the examples given bellow.
- Alahacoon, N.; Edirisinghe, M. Spatial Variability of Rainfall Trends in Sri Lanka from 1989 to 2019 as an Indication of Climate Change. ISPRS Int. J. Geo-Inf.2021, 10, 84. https://doi.org/10.3390/ijgi10020084
- Haylock, Malcolm R., Thomas C. Peterson, Lincoln M. Alves, Tércio Ambrizzi, Y. M. T. Anunciação, J. Baez, V. R. Barros et al. "Trends in total and extreme South American rainfall in 1960–2000 and links with sea surface temperature." Journal of climate19, no. 8 (2006): 1490-1512.
- Sabattini, Julian Alberto, and Rafael Alberto Sabattini. "Rainfall Trends in Humid Temperate Climate in South America: Possible Effects in Ecosystems of Espinal Ecoregion." (2021).
- Alahacoon, N.; Edirisinghe, M.; Simwanda, M.; Perera, E.; Nyirenda, V.R.; Ranagalage, M. Rainfall Variability and Trends over the African Continent Using TAMSAT Data (1983–2020): Towards Climate Change Resilience and Adaptation. Remote Sens.2022, 14, 96. https://doi.org/10.3390/rs14010096
- Endo, Nobuhiko, Jun Matsumoto, and Tun Lwin. "Trends in precipitation extremes over Southeast Asia." Sola5 (2009): 168-171.
Study area and data preparation (Please check the Journal format for the topic carefully).
Study Area
- Figure 3 Ledged is not appear well and need to improve the visibility of the figure.
- It is necessary to reduce the length of the study area by removing less important information as more information is provided for the study area.
Data
- The rainfall locations can be introduced in the same study area map and remove the Figure 4 from the paper.
- It is important to provide the details on the data gaps of the station data as most of time observe rainfall data having data gaps.
Methodology
- Lines 222 to 246 provide general information on climate change and its impact, and since it has already been discussed in the introductory section, my recommendation is to remove or move this section to the introductory section. It is more important for the authors to provide only details of the applied methodology in this section.
Results and discussion
Rainfall trend in Syria
- Authors need to provide the reasons why they selected only specific station to represent the rainfall trend results.
- All the figures in this section must be arrange correctly before publishing in a scientific journal and need to provide a good visibility for the figures (It difficult to read the letters in the charts). My recommendation is to provide all the figures on a single page with two columns. Current chart representation is not suitable for representation in scientific publication (it is better for a dissertation or an MSc dissertation).
- I have a lot of confusion as to why introducing the Man-Kendal test in the method section gives a linear trend in the result section. I strongly believe that Man-Kandel trend results should be better represented than simple linear trend analysis.
Man-Kandel results
- Figure 17 is difficult to read and need to improve it.
- In the description related to the Figure 17 – 19, it is not easy to understand what authors are trying to talk about the “Kendal Tau” variation over the time. It should be clearly address in the paper.
- Table 2: It need to explain what the parameters are explain in the table in the figure caption.
- Lines 500 – 514 needs to introduce in the methodology section as it is a part of the methodology.
- Figure 22 needs to be improved to represent the results. (Please use properly the general cartography).
Conclusion
Conclusion should be improved, and it should be providing the quantitative information of the study findings to strength the Authors conclusion.
Author Response
We would like to thank the reviewer for valuable comments that have certainly improved the quality of the paper. Please find enclosed the revised paper in which the and reviewer comments have been addressed. The reviewer's comments have been addressed in the same order as in the reviewer's report.
Abstract
It is very important to provide the quantitative observation of the study in the abstract to strengthening it.
Response: Thanks, the abstract has been modified and quantitative observations of the study have been added.
Introduction
- References are missing in Lines 51-52, 62-65, 66-68, 4-88,
Response: The references have been added at the end of each paragraph.
- The introduction is too lengthy, and it is important to reduce the length by highlighting important points related to the topic.
Response: The introduction has been revised and unnecessary parts have been removed. Also, new references (10) have been added to update the current references with more recent publications till 2022.
- It is important to provide the recent and similar studies in the introduction on the rainfall trend analysis in different geographical context of the world as some of the examples given bellow.
- Alahacoon, N.; Edirisinghe, M. Spatial Variability of Rainfall Trends in Sri Lanka from 1989 to 2019 as an Indication of Climate Change. ISPRS Int. J. Geo-Inf.2021, 10, 84. https://doi.org/10.3390/ijgi10020084
- Haylock, Malcolm R., Thomas C. Peterson, Lincoln M. Alves, Tércio Ambrizzi, Y. M. T. Anunciação, J. Baez, V. R. Barros et al. "Trends in total and extreme South American rainfall in 1960–2000 and links with sea surface temperature." Journal of climate19, no. 8 (2006): 1490-1512.
- Sabattini, Julian Alberto, and Rafael Alberto Sabattini. "Rainfall Trends in Humid Temperate Climate in South America: Possible Effects in Ecosystems of Espinal Ecoregion." (2021).
- Alahacoon, N.; Edirisinghe, M.; Simwanda, M.; Perera, E.; Nyirenda, V.R.; Ranagalage, M. Rainfall Variability and Trends over the African Continent Using TAMSAT Data (1983–2020): Towards Climate Change Resilience and Adaptation. Remote Sens.2022, 14, 96. https://doi.org/10.3390/rs14010096
- Endo, Nobuhiko, Jun Matsumoto, and Tun Lwin. "Trends in precipitation extremes over Southeast Asia." Sola5 (2009): 168-171.
Response: Thanks, all of these references have been added to the paper and list of references.
Study area and data preparation
(Please check the Journal format for the topic carefully).
Response: The Journal format has been check and this part has been revised.
Study Area
- Figure 3 Ledged is not appear well and need to improve the visibility of the figure.
Response: The figure has been modified.
- It is necessary to reduce the length of the study area by removing less important information as more information is provided for the study area.
Response: The length of the study area has been reduced and unnecessary data has been removed.
Data
- The rainfall locations can be introduced in the same study area map and remove the Figure 4 from the paper.
Response: The location map (figure 3) presents the location, main cities and boarders but the location of stations (Figure 4) includes different names and putting all together may not be clear in one map. So we prefer to keep in 2 figures.
- It is important to provide the details on the data gaps of the station data as most of time observe rainfall data having data gaps.
Response: The data were collected from 1991 to 2009, where the data were available and any missing data have been estimated by calculating the mean at the same station for the same month.
Methodology
- Lines 222 to 246 provide general information on climate change and its impact, and since it has already been discussed in the introductory section, my recommendation is to remove or move this section to the introductory section. It is more important for the authors to provide only details of the applied methodology in this section.
Response: Thanks, it has been removed.
Results and discussion
Rainfall trend in Syria
- Authors need to provide the reasons why they selected only specific station to represent the rainfall trend results.
Response: The rainfall trend has been done for all stations but we put only significant stations that have been selected based on the highest average rainfall to give significant change in the indexes.
- All the figures in this section must be arrange correctly before publishing in a scientific journal and need to provide a good visibility for the figures (It difficult to read the letters in the charts). My recommendation is to provide all the figures on a single page with two columns. Current chart representation is not suitable for representation in scientific publication (it is better for a dissertation or an MSc dissertation).
Response: Thanks, all the figures have been modified.
- I have a lot of confusion as to why introducing the Man-Kendal test in the method section gives a linear trend in the result section. I strongly believe that Man-Kandel trend results should be better represented than simple linear trend analysis.
Response: Thanks, Man-Kandel trend was already calculated for all stations but we presented the results of linear trend to give future insight for the decision makers for what may happen in the future based on the trend in previous years.
Man-Kandel results
- Figure 17 is difficult to read and need to improve it.
Response: The figure has been modified.
- In the description related to the Figure 17 - 19, it is not easy to understand what authors are trying to talk about the “Kendal Tau” variation over the time. It should be clearly address in the paper.
Response: The non-parametric Mann-Kendall trend test is one of the most frequent statistical approaches for analyzing time series datasets. In this section we presented the results of Man-Kendal at all stations for the period (1991-2009). It was calculated for months and periods. For months it was divided into 3 seasons (Figure 17, 18 and 19) but for summer months (June, July and August) there is no rainfall in the study area.
- Table 2: It needs to explain what the parameters are explain in the table in the figure caption.
Response: Excuse me, we put only one table but if you mean Table 1, the parameters are explained in the results discussion.
- Lines 500-514 needs to introduce in the methodology section as it is a part of the methodology.
Response: This part “Evaluation of rainfall trends” is a separate part to present the use of GIS for geographical distribution of rainfall trends which includes one equation and brief introduction to the presented maps.
- Figure 22 needs to be improved to represent the results. (Please use properly the general cartography).
Response: The figure has been improved.
Conclusion
Conclusion should be improved, and it should be providing the quantitative information of the study findings to strength the Authors conclusion.
Response: Thanks, the conclusion has been modified.